# PROMPT AND PARAMETER CO-OPTIMIZATION FOR LARGE LANGUAGE MODELS

**Xiaohe Bo**[1][*], **Rui Li**[1][*], **Zexu Sun**[1], **Quanyu Dai**[2]
**Zeyu Zhang**[1], **Zihang Tian**[1], **Xu Chen**[1][†], **Zhenhua Dong**[2]
[1]Gaoling School of Artificial Intelligence, Renmin University of China
[2]Huawei Technologies Ltd.
{xiaohe,lirui121200,zeyuzhang,zihangtian,xu.chen}@ruc.edu.cn,
sunzexu0826@gmail.com,{daiquanyu,dongzhenhua}@huawei.com

## ABSTRACT

Prompt optimization and fine-tuning are two major approaches to improve the performance of Large Language Models (LLMs). They enhance the capabilities of LLMs from complementary perspectives: the former through explicit natural language, and the latter through implicit parameter updates. However, prior work has typically studied them in isolation, leaving their synergistic potential largely underexplored. To bridge this gap, in this paper, we introduce MetaTuner, a novel framework that jointly integrates prompt optimization and fine-tuning for LLM training. Specifically, we introduce two neural networks to generate prompts and parameters, respectively, while allowing them to share a common bottom encoding layer to enable knowledge sharing. By the guidance of the final supervised signals, our framework is optimized to discover the optimal combinations between the prompts and parameters. Given that prompt learning involves discrete optimization while fine-tuning operates in a continuous parameter space, we design a supervised regularization loss to train our framework effectively. Extensive experiments across diverse benchmarks show that our method consistently outperforms the baselines. To benefit the research community, we have released our project at https://github.com/BoXiaohe/MetaTuner.

## 1 INTRODUCTION

Large language models (LLMs) have demonstrated remarkable success across a wide range of domains, including e-commerce (Jin et al., 2024; Peng et al., 2024; Li et al., 2024; Ren et al., 2024), education (Wang et al., 2024a; Ghimire et al., 2024; Chu et al., 2025), and social science (Wang et al., 2025; Mou et al., 2025; Tang et al., 2024; Piao et al., 2025). Unlike traditional machine learning models, whose capabilities primarily stem from parameter tuning, LLMs can enhance their performance through two complementary strategies: prompt optimization and fine-tuning.

Previously, prompt optimization and fine-tuning are often treated as independent research lines. For example, in the line of prompt optimization, current methods (Liu et al., 2025b; Austin & Chartock, 2024; Yang et al., 2024b; Choi et al., 2024) mainly focus on discovering or learning appropriate input context that can elicit the desired behavior from frozen language models. One group of works (e.g., OPRO (Yang et al., 2024a)) focuses on optimizing prompts with LLMs, whereas another group (e.g., RLPrompt (Deng et al., 2022), BPO (Cheng et al., 2024)) trains an auxiliary model for prompt generation. In the line of fine-tuning, existing approaches concentrate on better training algorithms to adjust the model parameters. Supervised Fine-Tuning (SFT) proposes to optimize the model with cross-entropy loss, while more recently, RL-based approaches such as RLHF (Li et al., 2023b) and DPO (Rafailov et al., 2023) are developed to optimize the model based on human feedback signals.

While the above models have shown promising results in various domains, both prompt optimization and fine-tuning have their own limitations. Specifically, prompt optimization typically focuses on

---

[*] Equal contribution.
[†] Corresponding author.

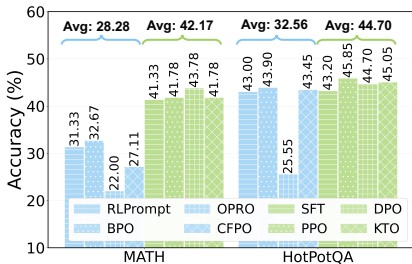
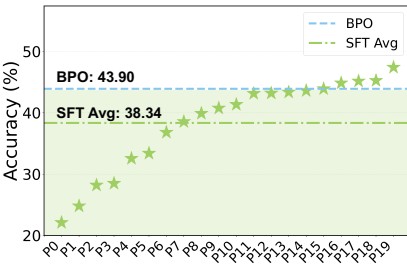

(a) Prompt optimization vs. fine-tuning.  (b) SFT with different prompts on HotpotQA.

Figure 1: Preliminary analysis of prompt optimization and fine-tuning methods. In the left subfigure, we present a comparison between four representative prompt optimization strategies and four popular fine-tuning methods. In the right subfigure, we utilize SFT as the backbone fine-tuning method and further evaluate the model with different training prompts, where the corresponding results are marked with stars. Additional details for the implementation can be found in the Appendix D.1.

refining general instructions within prompts to elicit the capabilities of LLMs. However, it often falls short in adapting the pretrained parameters to the complex patterns present in large-scale, task-specific datasets, especially when the knowledge encoded in the prompts and the model parameters may conflict. Fine-tuning, on the other hand, usually leverages manually crafted prompts as inputs during model training. However, the choice of input prompts can significantly affect fine-tuning outcomes, and existing methods offer no guarantee that these prompts are optimal for maximizing downstream performance. To further validate these statements, we conduct a series of preliminary experiments, and the corresponding results are presented in Figure 1. As illustrated in the left subfigure, by training the LLM parameters based on the datasets of MATH (Hendrycks et al., 2021) and HotpotQA (Yang et al., 2018), the fine-tuning methods substantially outperform the prompt optimization strategies on average. However, the effectiveness of fine-tuning is highly sensitive to the choice of prompts. As shown in the right subfigure, when sub-optimal prompts are used for training, the performance degrades and can even fall below that of prompt optimization methods. Motivated by the above mechanistic and empirical analyses, a natural question arises: **Can we design a unified framework to combine prompt optimization and fine-tuning to mitigate the weaknesses of each other?**

Answering the above question is not straightforward. To begin with, prompt optimization and fine-tuning have different execution processes and optimization objectives. The former optimizes the model externally by finding the suitable input context to activate existing knowledge, while the latter modifies the model internally by updating parameters to better fit the target data distribution. Effectively reconciling these differences to combine the two approaches requires careful design. In addition, prompt optimization is inherently a discrete optimization problem, whereas fine-tuning operates in a continuous optimization space. Combining the two, therefore, introduces mixed optimization challenges, such as non-differentiable gradients. Effectively leveraging the final supervision signal to address this discrete-continuous optimization problem remains a significant challenge.

To overcome the above challenges, we propose **MetaTuner**, a novel framework for joint prompt optimization and fine-tuning. Our approach first employs a meta encoder to encode the input query. The resulting embedding is then fed into two separate decoders—a prompt decoder and a parameter decoder—which simultaneously generate query-specific prompts and parameters for downstream large language models. This design allows the prompts and parameters to share the common knowledge from the meta encoder, promoting their mutual improvement. Considering that prompt learning is a discrete process while the parameter training is continuous, we design a supervised regularization loss to solve the mixed discrete-continuous optimization problem. Specifically, we first select the optimal rollout prompts to optimize the prompt decoder in a supervised learning manner, and then share the parameters of the prompt and parameter decoders for joint differentiable training.

The main contributions of this paper are summarized as follows: (1) We propose the idea of jointly optimizing the prompts and parameters of LLMs to leverage their complementary strengths and mitigate each other's weaknesses. (2) To realize this idea, we introduce a framework called MetaTuner, which unifies the prompt and parameter generation processes to enhance the overall task performance. (3) We design a supervised regularization objective to address the non-differentiability challenge and effectively optimize the MetaTuner framework. (4) We conduct extensive experiments on MATH, GSM8K, HotpotQA, and CosmosQA to demonstrate the effectiveness and robustness of our approach.

## 2 PROBLEM FORMULATION

In the post-training of LLMs, we typically work with an input-output pair $(x, y)$, where $x$ represents a question such as "What is the capital of France?", and $y$ corresponds to the expected answer like "Paris". Unlike traditional machine learning models, LLMs often require an additional prompt $p$ to rephrase the input $x$ in a way that better guides the model to generate the desired output $y$. In the example above, $p$ could be "Answer the following question briefly:". Let $\mathcal{M}$ be the language model, the overall process can be written as:

$$y = \mathcal{M}(p, x), \tag{1}$$

To enhance the capability of $\mathcal{M}$, there are two primary research directions: (1) Prompt optimization. The core idea of this approach is to find an optimal prompt $p$ that facilitates more accurate prediction of $y$. Given a loss function $\mathcal{L}$, the objective can be formulated as:

$$\min_{p_i} \sum_{i=1}^{N} \mathcal{L}(\mathcal{M}(p_i, x_i), y_i), \tag{2}$$

where $D = \{(x_i, y_i)\}_{i=1}^{N}$ denotes the set of input-output pairs. We use the subscript $i$ in $p_i$ to indicate that the prompt is input-sensitive. However, depending on the specific application, a shared or unified prompt across all inputs is also a natural extension. (2) Fine-tuning. This method regards the LLM as a traditional machine learning model, and optimizes its parameters based on the given input-output pairs. In specific, the objective is formulated as follows:

$$\min_{\theta} \sum_{i=1}^{N} \mathcal{L}(\mathcal{M}_{\theta}(p_i, x_i), y_i), \tag{3}$$

where we use $\theta$ to represent the parameters of the large language model.

Previous studies have employed various strategies (Li et al., 2023b; Rafailov et al., 2023; Ethayarajh et al., 2024; Yang et al., 2024a; Cheng et al., 2024; Deng et al., 2022; Liu et al., 2024a; Meng et al., 2024) to instantiate objectives (2) and (3), respectively. Here, we take a different perspective by formulating them within a unified framework, laying the foundation for letting prompt optimization and fine-tuning mutually enhance each other. Specifically, we regard the prompt as a "special parameter", then objectives (2) and (3) can be unified by the following loss function:

$$\min_{\theta, p_i} \sum_{i=1}^{N} \mathcal{L}(\mathcal{M}_{\theta}(p_i, x_i), y_i). \tag{4}$$

Compared with previous studies, this loss function considers prompts and model parameters as two complementary dimensions that jointly influence the prediction from $x_i$ to $y_i$. By optimizing them within a unified objective, we are actually finding the optimal combinations between the prompts and parameters to better fit specific tasks. The loss function can help to regularize any sub-optimal choices in either the prompt or parameter space. Readers may find that the above objective can also be applied to soft-prompting strategies (Liu et al., 2024b; 2022; Li & Liang, 2021; Lester et al., 2022; 2021). However, in this paper, we focus on generating explicit prompts, because the natural language prompts are more explainable, transparent, and human-friendly.

## 3 THE METATUNER FRAMEWORK

The overall architecture of our framework can be seen in Figure 2. We first design a neural network with continuous parameters to generate prompts, effectively transforming the discrete optimization problem into a continuous task. Next, we unify prompt optimization and fine-tuning by introducing a shared meta encoder that simultaneously produces input prompts and model parameters, allowing mutual adaptation to better fit the training samples. Finally, we design a supervised regularization loss to address the non-differentiable problem, enabling effective and stable framework optimization.

### 3.1 PROMPT GENERATION WITH CONTINUOUS NEURAL NETWORK

In practice, a prompt typically consists of dozens or even hundreds of words. Generating such a prompt entirely from scratch, without any reference, poses a significant challenge. To mitigate this

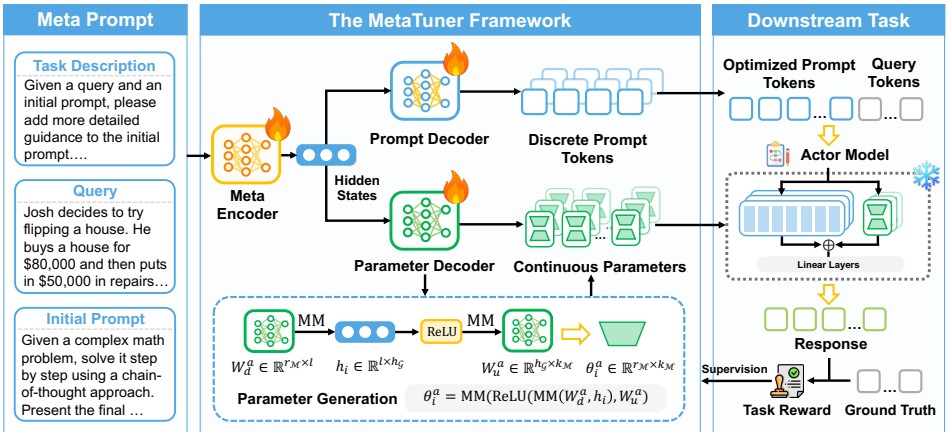

Figure 2: Illustration of the MetaTuner framework. The input query is first encoded by the meta encoder, and then two parallel decoders are utilized to generate the prompts and parameters, respectively, which are finally applied to the downstream actor model for problem solving.

difficulty, we first identify an initial prompt—albeit potentially sub-optimal—and then leverage a large language model (LLM) to rewrite it, thereby simplifying the prompt generation task. To reduce the effort of assigning an independent initial prompt to each input $x_i$, we let the LLM automatically explore the prompt space, starting from the same initial prompt. Formally, let $\mathcal{G}$ denote the model for prompt generation, then the rewritten prompt is represented as follows:

$$p_i = \mathcal{G}_\phi(\tilde{p}, x_i), \tag{5}$$

where we use $\phi$ to indicate the learnable parameters of $\mathcal{G}$, and $\tilde{p}$ is the predefined initial prompt. Based on equation (5), the objective (4) for LLM training is improved as follows:

$$\min_{\theta, \phi} \sum_{i=1}^{N} \mathcal{L}(\mathcal{M}_\theta(\mathcal{G}_\phi(\tilde{p}, x_i), x_i), y_i), \tag{6}$$

where the optimizable parameters $\theta$ and $\phi$ are all continuous.

## 3.2 JOINT PROMPT AND PARAMETER GENERATION

Intuitively, both prompts and model parameters aim to enhance the prediction of outputs $y_i$ from given inputs $x_i$, and are therefore expected to share common underlying knowledge. Furthermore, as outlined in the introduction, prompt optimization and fine-tuning each exhibit distinct limitations. This motivates the need for a principled approach that enables the two strategies to complement one another and mitigate each other's shortcomings.

To incorporate the above intuitions into our model design, we do not treat $\theta$ as independent parameters. Instead, we generate them using a neural network. This approach allows us to share parameters between the prompt generation and parameter generation models, thereby establishing a common foundation for both prompt optimization and fine-tuning. Formally, we partition the parameters of the prompt generation model $\mathcal{G}$ into two components: $\phi = \{\phi_s, \phi_p\}$. Here, $\phi_s$ denotes the shared parameters, which are also utilized for LLM parameter generation, while $\phi_p$ refers to the parameters exclusively used for prompt generation. We denote the model for LLM parameter generation as $\mathcal{F}$, and its parameters are also separated into two parts $\{\phi_s, \phi_q\}$, where $\phi_q$ is tailored to generating LLM parameters. Based on these formulation, we further improve objective (4) to obtain the final optimization target as follows:

$$\min_{\phi_s, \phi_p, \phi_q} \sum_{i=1}^{N} \mathcal{L}(\mathcal{M}_{\mathcal{F}_{(\phi_s, \phi_q)}(\tilde{p}, x_i)}(\mathcal{G}_{(\phi_s, \phi_p)}(\tilde{p}, x_i), x_i), y_i). \tag{7}$$

In this target, fine-tuning and prompt optimization are connected through a shared-private parameterization mechanism. On one hand, the shared parameters $\phi_s$ enable mutual regularization between the two approaches—any sub-optimal solution from one can be corrected by the other under the unified loss function. On the other hand, the private parameters $\phi_p$ and $\phi_q$ provide each approach with the flexibility to independently explore and identify its own optimal solutions.

### 3.3 Optimization with Supervised Regularization

In the above objective (7), the outputs of the prompt generation model $\mathcal{G}$ are discrete tokens, which makes it hard to directly compute the gradient of $\phi_p$. To address this challenge, we further introduce a surrogate loss based on supervised regularization as follows:

$$\min_{\phi_s, \phi_p, \phi_q} \sum_{(x_i, y_i) \in D_1} \mathcal{L}(\mathcal{M}_{\mathcal{F}_{(\phi_s, \phi_q)}(\tilde{p}, x_i)}(\mathcal{G}_{(\phi_s, \phi_p')}(\tilde{p}, x_i), x_i), y_i) + \sum_{(x_i, p_i) \in D_2} \alpha \cdot \mathcal{L}(\mathcal{G}_{(\phi_s, \phi_p)}(\tilde{p}, x_i), p_i). \tag{8}$$

In this objective, the first term is identical to (7), except that the private parameters used to generate the prompt (*i.e.*, $\phi_p'$) are kept fixed. The second term acts as a regularizer, optimizing $\phi_s$ and $\phi_p$ under a supervised learning paradigm. Here, $D_1$ denotes the original training dataset, while $D_2$ is an expert dataset in which each $p_i$ yields the correct $y_i$ given $x_i$. After several optimization batches, $\phi_p'$ is updated using $\phi_p$. In practice, one can optimize the above two terms either separately in sequence or jointly as a single loss. We evaluate both of these implementations in our experiments.

To address the above non-differentiable problem, we also explored strategies such as Gumbel-Softmax (Jang et al., 2017), but these methods generally produced inferior performance. We speculate that our supervised regularizer is more effective because it provides direct supervision on the optimal prompts, avoiding intermediate softening steps or approximations.

### 3.4 Framework Specification

In the above sections, we outlined the general design principles of our framework. In this section, we turn to the implementation details, with a particular focus on the key components $\mathcal{G}$ and $\mathcal{F}$.

**Specification of $\mathcal{G}$.** In our framework, an LLM such as Qwen2.5-7B (Team, 2024) is selected as the implementation of $\mathcal{G}$. In specific, we use the first $k$ decoder layers as the meta encoder (*i.e.*, $\phi_s$), while the subsequent layers serve as the prompt decoder (*i.e.*, $\phi_p$). We denote the hidden states by $h_i \in \mathbb{R}^{l \times h_{\mathcal{G}}}$, where $l$ represents the input sequence length and $h_{\mathcal{G}}$ corresponds to the hidden size.

**Specification of $\mathcal{F}$.** For training efficiency, we adopt LoRA (Hu et al., 2022) to fine-tune the downstream LLMs, with $\mathcal{F}$ responsible for generating the LoRA parameters. Specifically, let $\Delta W = \theta_i^b \cdot \theta_i^a$ denote the update matrix, where $\theta_i^b \in \mathbb{R}^{d_{\mathcal{M}} \times r_{\mathcal{M}}}$ and $\theta_i^a \in \mathbb{R}^{r_{\mathcal{M}} \times k_{\mathcal{M}}}$. $\mathcal{F}$ first share the same $\phi_s$ with $\mathcal{G}$ to produce $h_i$. After that, we derive $\theta_i^b$ and $\theta_i^a$ from the hidden states $h_i$ as follows:

$$\begin{aligned} \theta_i^b &= \text{MM}(\text{ReLU}(\text{MM}(W_d^b, h_i)), W_u^b), \\ \theta_i^a &= \text{MM}(\text{ReLU}(\text{MM}(W_d^a, h_i)), W_u^a), \end{aligned} \tag{9}$$

where $\phi_q = \{W_d^b \in \mathbb{R}^{d_{\mathcal{M}} \times l}, W_u^b \in \mathbb{R}^{h_{\mathcal{G}} \times r_{\mathcal{M}}}, W_d^a \in \mathbb{R}^{r_{\mathcal{M}} \times l}, W_u^a \in \mathbb{R}^{h_{\mathcal{G}} \times k_{\mathcal{M}}}\}$ is the parameter decoder. ReLU is the activation function, and MM denotes the matrix multiplication operation.

## 4 Experiments

In this section, we conduct extensive experiments to demonstrate the effectiveness of our framework, where we focus on the following research questions (**RQ**):

- **RQ1**: Can MetaTuner achieve better performance than existing state-of-the-art methods?
- **RQ2**: Do the different components of our framework contribute to the final performance?
- **RQ3**: Is the shared-private co-optimization structure in our framework effective?
- **RQ4**: Is the supervised regularization loss effective in enhancing model performance?
- **RQ5**: Can the capabilities of our framework generalize effectively to unseen datasets?
- **RQ6**: How does MetaTuner perform compared to the soft co-optimization approach?

In the following, we answer the above questions by reporting and analyzing the experiment results.

### 4.1 Experimental Setup

**Datasets and metrics.** We evaluate MetaTuner on four datasets: MATH (Hendrycks et al., 2021) and GSM8K (Cobbe et al., 2021) for mathematical reasoning, HotpotQA (Yang et al., 2018) and

Table 1: Comparing MetaTuner with baselines. We highlight the **best** and second-best results.

| Methods | Qwen2.5-7B | | | | Qwen2.5-3B | | | |
|---|---|---|---|---|---|---|---|---|
| | MATH | GSM8K | HotpotQA | CosmosQA | MATH | GSM8K | HotpotQA | CosmosQA |
| **Vanilla Methods** | | | | | | | | |
| Qwen2.5 | 18.44 | 51.63 | 19.85 | 36.80 | 17.11 | 49.36 | 3.95 | 25.00 |
| **Prompt Optimization Methods** | | | | | | | | |
| RLPrompt | 31.33 | 53.15 | 43.00 | 81.20 | 22.67 | 45.94 | 33.15 | 67.15 |
| BPO | 32.67 | 58.00 | 43.90 | 82.05 | 26.44 | 53.90 | 40.45 | 72.90 |
| OPRO | 22.00 | 75.06 | 25.55 | 69.10 | 21.00 | 62.62 | 10.55 | 69.35 |
| CFPO | 27.11 | 55.64 | 43.45 | 84.80 | 17.78 | 55.27 | 20.05 | 64.20 |
| **Fine-Tuning Methods** | | | | | | | | |
| SFT | 41.33 | 61.41 | 43.20 | 82.65 | 29.33 | 56.18 | 49.00 | 52.05 |
| PPO | 41.78 | 62.02 | 45.85 | 84.10 | 34.22 | 55.95 | 51.30 | 52.25 |
| DPO | 43.78 | 63.68 | 44.70 | 87.90 | 36.59 | 60.42 | 41.50 | 61.20 |
| KTO | 41.78 | 62.47 | 45.05 | 85.00 | 33.78 | 57.70 | 46.90 | 58.70 |
| **Hybrid Methods** | | | | | | | | |
| BetterTogether | 41.56 | 67.93 | 52.30 | 89.80 | 35.78 | 66.80 | 41.10 | 86.60 |
| MetaTuner-I | 48.22 | 78.54 | **55.75** | 92.15 | 40.89 | **73.46** | **59.05** | 86.55 |
| MetaTuner-J | **48.67** | **78.92** | 54.56 | **92.25** | **41.33** | 73.08 | 58.85 | **87.15** |

CosmosQA (Li et al., 2023a) for multi-hop and commonsense question answering. Each dataset is divided into training, development, and test sets, with details of the splits provided in the Appendix D.2. We report the exact match (EM) accuracy for MATH, GSM8K, and CosmosQA, while for HotpotQA, we adopt the F1 accuracy following the implementation of (Bo et al., 2024; Yao et al., 2024).

**Baselines.** We compare MetaTuner with a series of state-of-the-art approaches, including: (1) *Vanilla Methods:* Qwen2.5 (Team, 2024) is employed as an essential baseline to directly answer queries under the zero-shot setting. (2) *Prompt Optimization Methods:* RLPrompt (Deng et al., 2022), BPO (Cheng et al., 2024), OPRO (Yang et al., 2024a), and CFPO (Liu et al., 2025a). For RLPrompt and BPO, we use Qwen2.5 as the base prompt generator, while for LLM-based methods such as OPRO and CFPO, we use state-of-the-art proprietary LLMs as the prompt optimizer. (3) *Fine-Tuning Methods:* SFT, PPO (Ouyang et al., 2022), DPO (Rafailov et al., 2023), and KTO (Ethayarajh et al., 2024). For RL-based methods (PPO, DPO, and KTO), we first perform SFT as a warm-up stage and then construct alignment data following SPIN (Chen et al., 2024). (4) *Hybrid Methods:* BetterTogether (Soylu et al., 2024) is a hybrid method that iteratively optimizes the prompts and weights. However, it does not incorporate the parameter generation network and knowledge-sharing mechanism. As mentioned in the Section 3.2, we optimize our model with two strategies: In MetaTuner-I, we adopt a sequential optimization strategy, where the two terms in Equation 8 are optimized alternatively. In MetaTuner-J, we optimize Equation 8 in a unified manner.

**Implementation details.** We use `Qwen2.5-7B` and `Qwen2.5-3B` as the prompt generator $\mathcal{G}$ and the downstream actor model $\mathcal{M}$. Prior to joint training, we initiate an SFT stage to warm up $\mathcal{G}$ and $\mathcal{M}$ respectively. The training is carried out through Llama-Factory (Zheng et al., 2024), with the epoch set to 1, batch size set to 64, and the learning rate tuned in {1e-3, 5e-4, 1e-4, 5e-5}. Details of the stage can be found in the Appendix D.2. We then search the depth $k$ of $\phi_s$ in $\{K/4, K/2, 3K/4, K\}$ to determine the model structure, where $K$ denotes the number of decoder layers in $\mathcal{G}$. In the training process, we search $r_{\mathcal{M}}$ in {4, 8, 16, 32}, $\alpha$ in {0.1, 0.5, 0.9}, the temperature $t$ for online rollouts in {0, 0.3, 0.5, 0.7, 0.9}, and the learning rate in {1e-5, 5e-6, 1e-6, 5e-7}. Additionally, we introduce a hyper-parameter $\lambda$ to scale the generated LoRA before adding it to the original model weights, and search $\lambda$ over {0.01, 0.05, 0.1, 0.5}. During the inference stage, we employ greedy decoding to ensure reproducibility. All the experiments are conducted on eight 80G GPUs.

## 4.2 OVERALL PERFORMANCE

The overall comparison is reported in Table 1, where several observations can be drawn: (1) The hybrid optimization methods consistently outperform standalone prompt optimization and fine-tuning methods, highlighting the complementary strengths of prompt and parameter optimization. (2) Compared to BetterTogether, two variants of MetaTuner achieve an average relative improvement of 10.15% and 17.08% under the 7B and 3B backbone settings, respectively. This demonstrates

Table 2: Ablation Studies. We highlight the most and the second-most effective modules.

| Methods | Qwen2.5-7B | | | | Qwen2.5-3B | | | |
|---|---|---|---|---|---|---|---|---|
| | MATH | GSM8K | HotpotQA | CosmosQA | MATH | GSM8K | HotpotQA | CosmosQA |
| MetaTuner (w/o F) | 48.00 | 77.79 | 54.05 | 91.10 | 40.44 | 72.18 | 56.95 | 86.40 |
| MetaTuner (w/o P) | 46.22 | 78.54 | 53.90 | 91.00 | 39.78 | 72.25 | 57.65 | 86.55 |
| MetaTuner (w/o S) | 46.67 | 77.86 | 53.65 | 91.50 | 40.22 | 72.10 | 57.55 | 86.65 |
| MetaTuner | 48.67 | 78.92 | 54.56 | 92.25 | 41.33 | 73.08 | 58.85 | 87.15 |

that our design, featuring a shared meta encoder and a supervised regularization loss, captures the deeper synergies between prompts and parameters. (3) On most tasks, the MetaTuner-J variant yields slightly stronger results than the MetaTuner-I, suggesting that simultaneous optimization encourages consistent alignment between prompts and parameters. While on HotpotQA, the MetaTuner-I variant achieves better performance, with an average absolute improvement of 0.70% across the two backbone settings. This indicates that alternating updates may mitigate gradient interference between the two branches, leading to more stable training, which is particularly beneficial for challenging optimization scenarios. For subsequent analyses, we primarily focus on the MetaTuner-J variant.

## 4.3 ABLATION STUDIES

In the above section, we evaluate our model as a whole. In this section, we would like to see how different components in our model contribute to the final performance. In specific, we compare our final model with its three variants: In *MetaTuner (w/o F)*, we remove the fine-tuning component, that is, omitting the first term in Equation 8. In *MetaTuner (w/o P)*, we remove the prompt optimization component, that is, omitting the second term in Equation 8. In *MetaTuner (w/o S)*, we do not share the parameters between the prompt and parameter decoder, that is, we set $k = 0$ and equip $\mathcal{F}$ with an independent $\phi'_s$ with the same structure as $\mathcal{G}$. From the results in Table 2, we can see: Both terms in our supervised regularization loss play essential roles, since removing either of them leads to an average absolute accuracy drop of approximately 0.99% and 1.12%, respectively. These findings indicate that, rather than relying on the training of a single branch to search for the optimal prompt–parameter combination, jointly training both branches yields superior performance. Besides, instead of a simple combination of prompt optimization and fine-tuning, our framework enables two components to share common parameters in $\phi_s$ for mutual enhancement. The inferior performance of *MetaTuner (w/o S)* compared to the full model demonstrates the effectiveness of this design.

## 4.4 EFFECTIVENESS OF THE SHARED-PRIVATE CO-OPTIMIZATION STRUCTURE

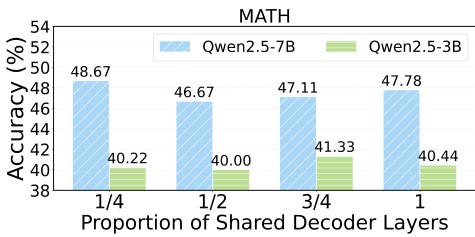

Figure 3: Performance with different proportions of shared decoder layers.

As demonstrated in the above experiments, parameter sharing between the fine-tuning and prompt optimization components is important for performance improvement. To further study the relation between the amount of shared parameters and the model performance, in this section, we examine how performance varies with different values of the sharing depth $k$ (introduced in Section 3.4), where $k$ is set to $\{K/4, K/2, 3K/4, K\}$ and $K$ denotes the maximum number of decoder layers in $\mathcal{G}$. From the results in Figure 3, we can observe that for the 7B model, a smaller sharing ratio (K/4) yields the best results, whereas for the 3B model, a larger sharing ratio (3K/4) performs better. This highlights the trade-off between the shared and private parameters: for the larger 7B model, its representation capacity is sufficiently strong, and excessive sharing may restrict the specialization of the prompt and parameter branches. Preserving more private layers therefore allows the model to better exploit its potential. In contrast, for the smaller 3B model, a higher sharing ratio strengthens representational consistency between the two branches, thereby improving training stability and downstream performance.

## 4.5 EFFECTIVENESS OF THE SUPERVISED REGULARIZATION LOSS

In our framework, a key design for addressing the mixed discrete–continuous optimization problem is the incorporation of the supervised regularizer in the objective (8). To evaluate its effectiveness, we

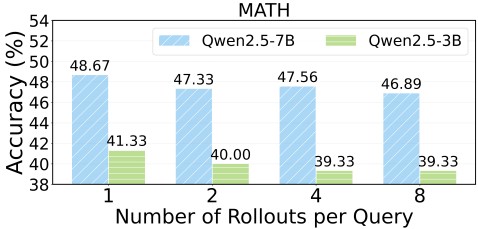

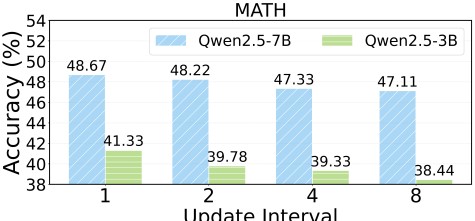

Figure 6: Influence of the rollout number. | Figure 7: Influence of the update frequency.

conduct comprehensive experiments based on four research questions: (1) Is our method superior to other well-known differentiable strategies such as Gumbel-Softmax? (2) In selecting the dataset $D_2$, should it be derived from the optimized model itself or from a stronger model? (3) How does the number of rollout samples affect performance? (4) How does the update frequency of $\phi_p$ influence the final performance? In the following, we report and analyze the experiment results.

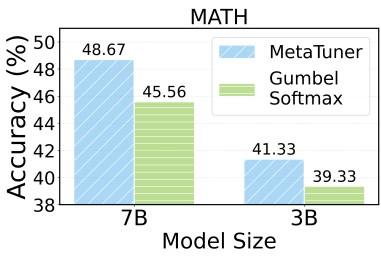

Figure 4: Comparison results between our method with Gumbel-Softmax.

**Comparing with other differentiable strategies**. We compare our method with Gumbel-Softmax, where the categorical prompt token sampling is replaced with a Gumbel-Softmax distribution to enable differentiable sampling. As shown in Figure 4, our method significantly outperforms Gumbel-Softmax. The main reason is that Gumbel-Softmax relies on continuous relaxation to approximate the discrete sampling. Although it ensures differentiability in theory, such softening operations introduce gradient bias, so that the generated prompts may deviate semantically from the truly optimal discrete solutions. In contrast, our methods directly optimize over the discrete action space, avoiding potential approximation errors.

**Comparison between different generation methods of** $D_2$. We compare the performances by generating $D_2$ using the proprietary expert LLM or the optimized model itself. As shown in Figure 5, while the expert method improves training efficiency, our model achieves better performance. The key reasons are threefold: first, our model ensures distribution alignment by generating data directly from the current policy, thereby reducing distribution shift between training and inference; second, it provides more targeted feedback, as rewards are based on the model's own outputs and directly address its weaknesses, unlike expert samples that may overlook error patterns; third, our model enables

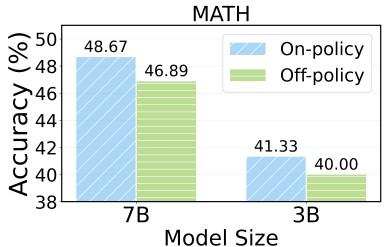

Figure 5: Comparison results between different methods for generating $D_2$.

stronger exploration, allowing the policy to continuously discover new prompts and iteratively refine itself through task feedback, whereas the expert data remain static and limit long-term improvement.

**Influence of the number of rollout samples**. When building $D_2$, an important factor is how many rollout samples should be generated for each query. In this section, we study the relation between this factor and the model performance. The results in Figure 6 show that as the number of rollout samples increases, the performance of both model sizes declines. On one hand, over-exploration may cause the model to frequently adjust its strategy, and these adjustments may not lead to short-term performance improvements, potentially interfering with valuable information the model has already learned. On the other hand, excessive exploration can cause overfitting, making the model overly sensitive to noise or irrelevant features in the training data, which ultimately degrades the downstream performance. Therefore, the number of rollout samples should be carefully adjusted according to the specific task and model characteristics to avoid the negative impact of excessive exploration.

**Influence of the update frequency of** $\phi_p$. In the first term of our objective (8), the private parameters $\phi_p$ for prompt generation is updated in an asynchronous manner to ensure that the objective is fully differentiable. Here, we investigate the influence of different update frequencies and report the results in Figure 7. On the horizontal axis, the values indicate how many training steps elapse before $\phi_p$ used for rollout sampling is replaced with the most recently trained one. The results on both model sizes

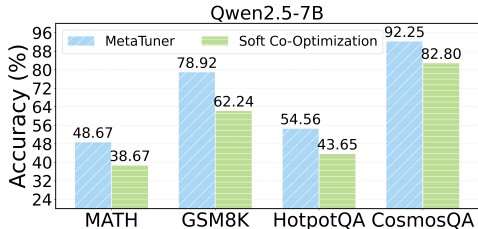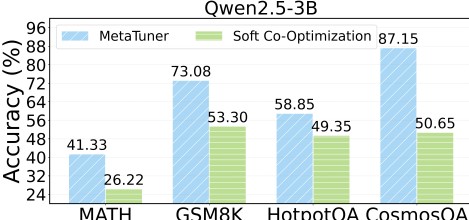

Figure 9: Comparing MetaTuner with Soft Co-Optimization method.

consistently show that sparser updates lead to poorer performance, since frequent updates allow the sampled prompts to be closely aligned with the current actor model, thereby providing more effective optimization signals. Moreover, we observe that the 3B model exhibits a greater performance drop, which suggests that smaller models are more sensitive to the timeliness of generated prompts, whereas the 7B model, with its stronger representational capacity, is more resilient to such lag effects.

## 4.6 GENERALIZATION ANALYSIS

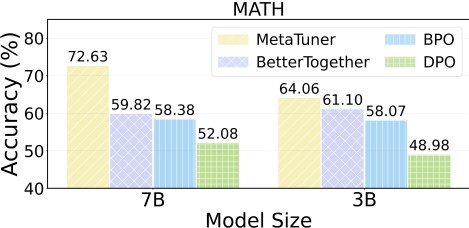

Figure 8: Generalization experiment.

In the above sections, we evaluate our model by training and testing on the same dataset. To further study the generalization capability of our method, we conduct out-of-distribution experiments by training the model on MATH, HotpotQA, and CosmosQA, and evaluate it on GSM8K. For comparison, we select DPO, BPO, and BetterTogether as the baselines. From the results in Figure 8, we can see: MetaTuner outperforms all other methods, confirming the advantage of joint optimization of prompts and parameters in enhancing the generalization ability. Compared to BetterTogether, MetaTuner achieves better synergistic optimization, thereby adapting more effectively to complex and unseen tasks. DPO, as a fine-tuning method, although effective in parameter adjustment, lacks support in prompt selection, while BPO, which relies solely on prompt optimization and fails to fully leverage the model's capacity, both lead to poorer generalization performance.

## 4.7 COMPARISON WITH SOFT CO-OPTIMIZATION METHOD

We compare MetaTuner with soft co-optimization methods and report the results in Figure 9. We employ prefix tuning (Li & Liang, 2021) for soft prompt optimization, while using LoRA as the fine-tuning approach, and fine-tune the model with SFT. From the results, we can find that the optimization of discrete prompts is more stable. Existing work (Bhardwaj et al., 2022; Razdaibiedina et al., 2023) has shown that soft prompt tuning can suffer from "representation collapse", where prompts for different inputs converge to nearly identical vectors. In MetaTuner, we optimize discrete prompts, i.e., prompts must take the form of readable natural language, with each token subject to clear semantic constraints. This forces the optimization process to search over structured, human-interpretable instructions rather than arbitrary continuous vectors, thereby reducing the risk of overfitting.

## 5 RELATED WORK

**Fine-tuning.** Fine-tuning is a standard approach for adapting pre-trained language models to specific tasks. Recent advances in parameter-efficient fine-tuning (PEFT) scale this to LLMs by training small plug-in modules under a frozen backbone. Representative methods include Adapter Tuning (Houlsby et al., 2019), which inserts lightweight adapter layers, and LoRA (Hu et al., 2022), which applies low-rank updates to weight matrices. Building on these, recent works propose various post-training strategies (Ethayarajh et al., 2024; Guo et al., 2024; Jung et al., 2024; Xu et al., 2024; Yang et al., 2024a). For instance, RLHF (Li et al., 2023b) incorporates human feedback to improve alignment, while DPO (Rafailov et al., 2023) streamlines preference modeling through contrastive learning.

**Prompt optimization.** Prompt optimization has become a research hotspot with the rise of in-context learning. Early works focus on continuous prompt tuning (Li & Liang, 2021; Lester et al., 2022; Liu

et al., 2024b; 2022; Lester et al., 2021), where learnable vectors are attached to the input. To mitigate the interpretability challenge (Wen et al., 2023), recent studies turn to discrete prompt optimization, which generates prompts in natural language. Existing approaches fall into two categories. The first employs LLMs as prompt optimizers (Liu et al., 2025b), typically in a feedback-driven manner, via strategies such as reflection-based refinement (Yang et al., 2024a;b; Austin & Chartock, 2024) or Monte Carlo Tree Search (MCTS) (Wang et al., 2024b; Pryzant et al., 2023; Liu et al., 2025a). The second (Kong et al., 2024; Juneja et al., 2024; Choi et al., 2024; Wang et al., 2024b; Chen et al., 2023) trains an auxiliary model to generate better prompts, most often with reinforcement learning, since prompts lack explicit ground truth and can be naturally guided by the task rewards.

**Joint optimization.** Recently, BetterTogether (Soylu et al., 2024) proposes to update prompts and model weights together to improve the task performance. In BetterTogether, prompt optimization and fine-tuning correspond to discrete and continuous optimization problems, respectively. During the prompt optimization phase, it samples candidate demonstration sets from the training data, evaluates them on a validation set to select the best few-shot prompt. In contrast, the fine-tuning phase is formulated as a continuous optimization problem, where supervised fine-tuning is applied to update the model parameters for better task adaptation. Different from BetterTogether, we first transform the discrete–continuous optimization into a fully continuous problem, then propose a knowledge sharing mechanism and a supervised regularization loss to enable effective end-to-end co-optimization.

## 6 CONCLUSION

In this paper, we propose MetaTuner, a novel framework that effectively combines prompt optimization and fine-tuning to enhance the performance of LLMs. We first introduce a prompt generator to transform the discrete–continuous optimization problem into a fully continuous one, and then design a dual-branch framework with a shared meta encoder to generate prompts and parameters simultaneously. We further propose a supervised regularization loss to address the non-differentiability challenge, thereby enabling stable end-to-end training of the framework. Extensive experiments on four benchmarks demonstrate the effectiveness and robustness of MetaTuner for joint optimization.

## ACKNOWLEDGEMENT

This work is supported in part by National Natural Science Foundation of China (No. 62472427 and No. 62422215), Major Innovation & Planning Inter-disciplinary Platform for the "DoubleFirst Class" Initiative, Renmin University of China, Public Computing Cloud, Renmin University of China, fund for building world-class universities (disciplines) of Renmin University of China.

## REPRODUCIBILITY STATEMENT

We take several steps to ensure the reproducibility of our work. Firstly, we detail the joint optimization objective and supervised regularization loss in Sections 3.2 and 3.3, and present the architectural details in Section 3.4. These descriptions are intended to make both the methodological design and implementation choices transparent. Secondly, we document the experimental setup in Section 4.1, including datasets, evaluation metrics, and training configurations, to enable the replication of the training and evaluation procedures. We further elaborate on data splits, the warm-up stage, and the meta-prompt design in the Appendix D.2 to facilitate replication. We believe these efforts provide the community with sufficient resources to reproduce our work and extend it in future research.

## ETHICS STATEMENT

The development and deployment of frameworks like MetaTuner, which aim to enhance the performance of large language models through joint prompt-parameter optimization, raise several ethical considerations. First, as with any system based on large-scale language models, there is a risk of inheriting or amplifying biases present in the training data. If not carefully addressed, the improved adaptability of MetaTuner could inadvertently reinforce harmful stereotypes or generate misleading information. Second, the computational resources required to train and fine-tune large language models contribute to environmental concerns related to energy consumption and carbon footprint. Future work should explore new ways to further reduce the resource demands.

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

Table 3: Main results with more datasets and base models.

| Dataset | Checkmate in One Move | | | MATH | | |
|---|---|---|---|---|---|---|
| Base Model | Qwen2.5-3B | Qwen2.5-7B | Llama3.2-3B | Llama3.1-8B | Llama3.2-3B | Mistral-7B |
| Vanilla Methods | | | | | | |
| Base Model | 0.00 | 0.00 | 0.00 | 21.33 | 25.11 | 10.89 |
| Prompt Optimization Methods | | | | | | |
| RLPrompt | 0.00 | 0.00 | 0.00 | 24.00 | 25.56 | 12.00 |
| BPO | 0.00 | 0.86 | 1.43 | 26.22 | 26.22 | 12.44 |
| OPRO | 0.00 | 0.57 | 1.14 | 25.77 | 26.22 | 11.78 |
| CFPO | 0.00 | 0.00 | 1.43 | 25.33 | 27.11 | 11.56 |
| Fine-Tuning Methods | | | | | | |
| SFT | 13.71 | 18.00 | 18.85 | 24.44 | 33.11 | 14.00 |
| PPO | 14.00 | 18.29 | 19.43 | 23.78 | 32.22 | 14.67 |
| DPO | 13.42 | 18.86 | 19.14 | 26.44 | 34.00 | 16.00 |
| KTO | 14.57 | 18.57 | 20.00 | 29.78 | 33.56 | 13.78 |
| Hybrid Methods | | | | | | |
| BetterTogether | 15.14 | 19.71 | 20.57 | 30.44 | 34.44 | 16.44 |
| MetaTuner-J | **17.71** | **20.86** | **21.43** | **31.78** | **18.44** | **35.78** |

# A  LLM USAGE

The usage of LLMs in this work includes two aspects: (1) Writing assistance. We use LLMs to improve the fluency and clarity of certain parts of the manuscript. All generated text is carefully reviewed, revised, and validated by the authors to ensure accuracy. (2) Dataset construction. We employ LLMs as auxiliary tools to generate candidate prompts as described in the Appendix D.2. After task-reward filtering, the collected data are used to improve the open-source smaller models' ability to follow prompt-rewriting instructions. All scientific claims, methodological designs, implementations, analyses, and conclusions are original contributions of the authors. LLMs are used solely as supporting tools, and the authors take full responsibility for the integrity and accuracy of the work.

# B  LIMITATIONS

While MetaTuner presents a promising framework for jointly optimizing prompts and parameters, there are several limitations that point to potential directions for future research. First, the RL-based approach used to address the non-differentiability issue in prompt optimization relies heavily on task reward signals (e.g., 0/1 accuracy). Such signals can be overly sparse in certain tasks, which may hinder optimization efficiency. Future work could explore more fine-grained reward designs, such as model uncertainty, to provide richer and more informative optimization signals. Second, MetaTuner generates a dedicated set of prompt and parameters for each query, which may face computational challenges in resource-constrained environments. A promising direction is to integrate the Mixture of Experts (MoE) paradigm: generate and maintain a fixed pool of expert LoRA modules, and dynamically route each query to the most suitable expert(s). This design could avoid generating a new LoRA for every input, thereby improving the efficiency and scalability.

# C  ADDITIONAL EXPERIMENTS

## C.1  MAIN RESULTS WITH MORE DATASETS AND BASE MODELS

To further validate the effectiveness of MetaTuner across broader scenarios and model architectures, we conduct additional experiments on the `Checkmate in One Move` dataset using multiple base models, including Llama3.2-3B (`Llama-3.2-3B-Instruct`), Llama3.1-8B (`Llama-3.1-8B-Instruct`), and Mistral-7B (`Mistral-7B-Instruct`). The results in Table 3 show that MetaTuner consistently outperforms baseline methods, demonstrating the efficacy of our proposed prompt–parameter co-optimization framework, which is built upon parameter sharing and a supervised regularization loss.

## C.2 Additional Generalization Analysis

We provide additional OOD experiments in Figure 10. The results show that MetaTuner achieves better out-of-distribution generalization compared to baseline methods. Theoretically, the generalization capability of our framework stems from the representation sharing mechanism. In MetaTuner, we can regard prompt generation and parameter generation as two distinct tasks and enforce them to share a common meta encoder. This shared-private multi-head architecture inherently induces a regularization effect (Park et al., 2019; Wang et al., 2022): the shared encoder must simultaneously support the learning objectives of multiple heads, compelling it to capture stable, core structural features that are invariant across tasks and independent of the specific training distribution. This aligns with classical theoretical results in multi-task learning (MTL) (Baxter, 2000; Maurer et al., 2016; D'Eramo et al., 2024; Bairaktari et al., 2023; Zhang et al., 2025): When multiple tasks share intermediate representations, the effective hypothesis space of the model is constrained, leading to tighter generalization error bounds. Intuitively, the model cannot learn highly task-specific, high-capacity representations independently for each task; instead, all tasks must find compatible representations within the same shared feature space. This capacity constraint significantly reduces overfitting to the idiosyncrasies of the training data distribution, thereby enhancing generalization across tasks and datasets. We believe that existing theoretical works on multi-task learning provide a solid foundation for understanding the generalization properties of MetaTuner.

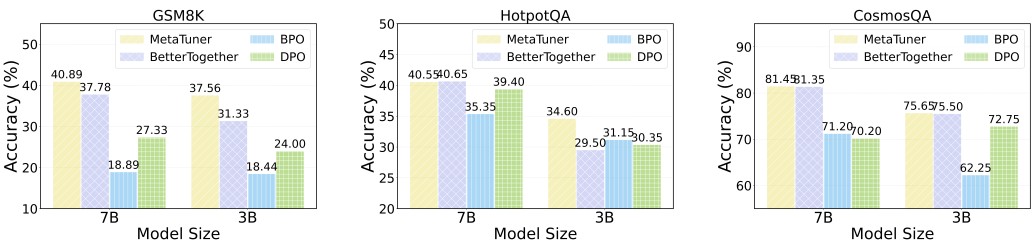

Figure 10: Additional generalization results. We select one dataset from the four (MATH, GSM8K, HotpotQA, CosmosQA) as the test set and use the remaining three as the training set.

## C.3 Case Study

**Joint optimization for superior reasoning.** We present a case from the MATH dataset in Figure 11 to illustrate the advantages of MetaTuner over existing methods. Compared with pure prompt optimization approaches such as BPO, which can produce step-by-step reasoning but often generate overly lengthy intermediate steps and occasionally commit calculation errors that lead to incorrect answers, MetaTuner produces reasoning traces that are both accurate and concise while strictly adhering to the required output format. This improvement stems from its joint optimization of prompts and parameters, which allows the model to better adapt to downstream tasks. In contrast, when compared with traditional fine-tuning approaches such as DPO, MetaTuner also demonstrates clear advantages. Although DPO can arrive at the correct numerical solution, it fails to comply with formatting requirements and tends to generate relatively rigid and less context-aware reasoning. MetaTuner, by dynamically generating high-quality prompts that activate relevant internal knowledge while simultaneously refining model parameters, effectively combines the strengths of both prompt optimization and fine-tuning. As a result, MetaTuner achieves more stable, flexible, and diverse reasoning patterns, demonstrating both correctness and expressiveness.

**Learned prompts for different base models.** Below, we provide several examples in Table 4 to analyze the commonalities and differences of prompts learned for different base models. As shown in the examples, both models (Qwen2.5-3B and Qwen2.5-7B) retain the core structure of the initial prompt (e.g., "step-by-step reasoning," "answer in \boxed{}", and "word limit"), indicating that MetaTuner can stably optimize instruction formats that respect task constraints across different models. However, the 7B model tends to generate more task-specific and procedurally explicit instructions (e.g., for geometry problems, explicitly instructing "first compute the new dimensions, then calculate the area change"), whereas the 3B model favors more general guidance (e.g., "explain the relevant mathematical concepts" or "include a table or diagram"). We hypothesize that the

**Question:** If $f(x) = \frac{3x-2}{x-2}$, what is the value of $f(-2) + f(-1) + f(0)$? Express your answer as a common fraction.

**Ground truth:** $f(-2) + f(-1) + f(0) = \frac{(3(-2)-2)}{(-2-2)} + \frac{3(-1)-2}{(-1-2)} + \frac{3(0)-2}{(0-2)} = \frac{-8}{-4} + \frac{-5}{-3} + \frac{-2}{-2} = 2 + \frac{5}{3} + 1 = \boxed{\frac{14}{3}}$.

### Prompt Optimization-BPO

**Input:** Solve complex mathematical problems step by step using a chain-of-thought approach. Keep the explanation concise. Present the final result in LaTeX using a '\boxed{}'. Ensure the response is correct, helpful, and harmless. Question: {question}

**Output:** To solve this problem, we need to evaluate the function $f(x)$ at $x = -2, x = -1$, and $x = 0$, and then add the results together.

First, let's evaluate $f(-2)$: $f(-2) = \frac{3(-2)-2}{-2-2} = \frac{-6-2}{-4} = \frac{-8}{-4} = 2$

Next, let's evaluate $f(-1)$: $f(-1) = \frac{3(-1)-2}{-1-2} = \frac{-3-2}{-3} = \frac{-5}{-3} = \frac{5}{3}$

Finally, let's evaluate $f(0)$ : $f(0) = \frac{3(0)-2}{0-2} = \frac{-2}{-2} = 1$

Now, let's add the results together: $f(-2) + f(-1) + f(0)$

$= 2 + \frac{5}{3} + 1 = \frac{6}{3} + \frac{5}{3} + \frac{3}{3} = \frac{14}{3}$ .

### Fine-tuning-DPO

**Input:** Question: {question}

**Output:** To find the value of $(f(-2) + f(-1) + f(0))$ for the function $f(x) = \frac{3x-2}{x-2}$, we need to evaluate the function at $(x = -2)$, $(x = -1)$, and $(x = 0)$ and then sum the results.

First, let's evaluate $(f(-2))$: $f(-2) = \frac{3(-2)-2}{-2-2} = \frac{-6-2}{-4} = \frac{-8}{-4} = 2$

Next, let's evaluate $(f(-1))$: $f(-1) = \frac{3(-1)-2}{-1-2} = \frac{-3-2}{-3} = \frac{-5}{-3} = \frac{5}{3}$

Finally, let's evaluate $(f(0))$: $f(0) = \frac{3(0)-2}{0-2} = \frac{-2}{-2} = 1$

### MetaTuner

**Input:** Given a complex mathematical problem, solve it step by step using a chain-of-thought approach. First, evaluate the function $f(x) = \frac{3x-2}{x-2}$ at the specified points: $x = -2, x = -1$, and $x = 0$. Then, calculate the sum of these values. Ensure clarity in each step of the calculation. The answer and reasoning process should be concise. Present the final result in LaTeX using a '\boxed{}' without any units. Question: {question}

**Output:** We have $f(-2) = \frac{3(-2)-2}{-2-2} = \frac{-8}{-4} = 2$, $f(-1) = \frac{3(-1)-2}{-1-2} = \frac{-5}{-3} = \frac{5}{3}$, and $f(0) = \frac{3(0)-2}{0-2} = \frac{-2}{-2} = 1$. Thus, $f(-2) + f(-1) + f(0) = 2 + \frac{5}{3} + 1 = \boxed{\frac{14}{3}}$.

Figure 11: Case study from the MATH dataset.

difference is primarily due to the varying task-solving capabilities of the base models. For the weaker 3B model, overly specific instructions risk embedding subtle errors that can mislead inference. MetaTuner thus learns to generate more robust, general prompts for smaller models. In contrast, the more capable 7B model can reliably follow and benefit from detailed, task-specific guidance, leading MetaTuner to produce finer-grained instructions for it.

**Learned prompts across different tasks.** We provide several cases in Table 5 to demonstrate the learned prompts across different tasks. From the examples, MetaTuner's learned prompts exhibit clear cross-task commonalities and transferable patterns. In MATH and GSM8K, both learned prompts refine the initial CoT instruction by explicitly guiding step decomposition, intermediate computation, and clear explanation—despite differing problem types. In HotpotQA, the prompt preserves the core principle of concise, evidence-based answering, and sharpened to extract only task-relevant details while avoiding redundancy. In CosmosQA, the prompt emphasizes grounding the answer in contextual key details. In summary, despite differences in task-specific requirements, the prompts learned by MetaTuner consistently promote clear, logically structured responses and adapt their instructional details according to the nature of each task. This indicates that MetaTuner is able to identify and leverage shared structural patterns across tasks, while tailoring guidance to individual task needs, thereby enhances the transferability and adaptability of its prompts across diverse settings.

Table 4: Cases of learned prompts with different base models.

| | Initial Prompt |
|---|---|
| | Given a complex mathematical problem, solve it step by step using a chain-of-thought approach. The answer and reasoning process should not exceed 250 words. Present the final result in LaTeX using a "\boxed{}" without any units. |
| | **Case 1** |
| Query | The lengths of two opposite sides of a square are decreased by 40% while the lengths of the other two sides are increased by 50% to form a rectangle. By what percent does the square's area decrease? |
| Learned Prompt (Qwen2.5-3B) | Given a complex mathematical problem, solve it step by step using a chain-of-thought approach. The answer and reasoning process should not exceed 250 words. Present the final result in LaTeX using a "\boxed{}" without any units. Additionally, provide a detailed explanation of the mathematical concepts involved in the problem, such as the properties of squares and rectangles, and the relationship between their areas. The explanation should be concise and clear. |
| Learned Prompt (Qwen2.5-7B) | Given a complex mathematical problem involving geometric transformations, solve it step by step using a chain-of-thought approach. First, calculate the new dimensions of the rectangle formed by decreasing and increasing the sides of the square. Then, determine the original and new areas of the square and rectangle. Finally, calculate the percentage decrease in the square's area. The answer and reasoning process should not exceed 250 words. Present the final result in LaTeX using a "\boxed{}" without any units. |
| | **Case 2** |
| Query | Find the greatest integer less than $(\sqrt{7} + \sqrt{5})^6$. (Do not use a calculator!) |
| Learned Prompt (Qwen2.5-3B) | Given a complex mathematical problem, solve it step by step using a chain-of-thought approach. The answer and reasoning process should not exceed 250 words. Present the final result in LaTeX using a "\boxed{}" without any units. Provide a detailed explanation of the mathematical concepts and techniques used in the solution, including any relevant formulas or theorems. The prompt should also include a table or diagram to illustrate the problem and its solution. |
| Learned Prompt (Qwen2.5-7B) | Given a complex mathematical problem, solve it step by step using a chain-of-thought approach. Begin by estimating the values of the square roots involved and then calculate the expression $(\sqrt{7} + \sqrt{5})^6$. Use algebraic identities and approximations to simplify the process. Ensure that each step is clearly explained and logically follows from the previous one. The answer and reasoning process should not exceed 250 words. Present the final result in LaTeX using a "\boxed{}" without any units. |

Table 5: Cases of learned prompts with Qwen2.5-7B across different tasks.

| | MATH |
|---|---|
| Query | The lengths of two opposite sides of a square are decreased by 40% while the lengths of the other two sides are increased by 50% to form a rectangle. By what percent does the square's area decrease? |

Table 5: Cases of learned prompts with Qwen2.5-7B across different tasks. (Continued)

| | |
|---|---|
| Initial Prompt | Given a complex mathematical problem, solve it step by step using a chain-of-thought approach. The answer and reasoning process should not exceed 250 words. Present the final result in LaTeX using a "\boxed{}" without any units. |
| Learned Prompt | Given a complex mathematical problem involving geometric transformations, solve it step by step using a chain-of-thought approach. First, calculate the new dimensions of the rectangle formed by decreasing and increasing the sides of the square. Then, determine the original and new areas of the square and rectangle. Finally, calculate the percentage decrease in the square's area. The answer and reasoning process should not exceed 250 words. Present the final result in LaTeX using a "\boxed{}" without any units. |

**GSM8K**

| | |
|---|---|
| Query | Ted the T-Rex was planning to bring potato salad to the dinosaur picnic. He knows that an adult dinosaur will eat 10 lbs of potato salad, and a child will eat half as much as an adult. If there will be 20 adults and 5 children at the picnic, how many pounds of potato salad does Ted need to bring to the picnic if he hopes to have enough to feed everyone? |
| Initial Prompt | Given a complex mathematical problem, solve it step by step using a chain-of-thought approach. |
| Learned Prompt | Solve the following complex mathematical problem step by step using a chain-of-thought approach. Break down the problem into smaller parts, calculate intermediate values, and provide a clear explanation for each step. |

**CosmosQA**

| | |
|---|---|
| Query | The black cloud's clinic is full of complex patients who need three referrals apiece. The white cloud's clinic is full of patients who don't show up. What finally got me to believe in the power of the black cloud was one resident I worked with last year. One month she was working in the clinic but covering call in the PICU every fourth night (as I am doing this month). Why are you cover call in the PICU this month? A. The schedule for that job rotates monthly and new people have the responsibility of doing it monthly B. I am the best at covering calls at the PICU so I have that job C. None of the above choices. D. I needed to make extra money so I applied for the position |
| Initial Prompt | Based on the provided context, select the most relevant and accurate option as the correct answer to the question. The answer should be directly related to the context, and must focus on the key details or events described. |
| Learned Prompt | Based on the provided context, select the most relevant and accurate option as the correct answer to the question. Consider the details about the rotation of responsibilities and the nature of the job in the PICU. The answer should be directly related to the context, focusing on the key details or events described. |

**HotpotQA**

Table 5: Cases of learned prompts with Qwen2.5-7B across different tasks. (Continued)

| | |
|---|---|
| Query | (1) UKF is a brand owned by Luke Hood and AEI Media that focuses on sharing all aspects of electronic music. UKF Music was created by Luke Hood, who began sharing bass music through his original two YouTube Channels, UKF Drum & Bass and UKF Dubstep in 2009. On 28 June 2012, UKF celebrated one billion YouTube views since its creation. The UKF brand now has five YouTube Channels: UKF Music, UKF Dubstep, UKF Drum & Bass, UKF Mixes, and UKF Live. UKF has expanded beyond its YouTube Channels to creating compilation series and podcasts, organizing events, offering merchandise as well as its own ticketing platform. 
 (2) Ysabella Brave is an American YouTube personality, artist, vocalist, singer and songwriter signed by Cordless Recordings, a division of the Warner Music Group. She was discovered through the popularity of her YouTube channels, Ysabella Brave and ysabellabravetalk. 
 (3) Steven Jay Williams (born July 24, 1974), also known by his online alias Boogie2988 or just Boogie, is an American YouTube personality best known for his video rants about video games and nerd culture as a character named "Francis". The Francis character is based on stereotypes of video game players and often parodies trending video game news, reaction and culture. Williams based the character on his early life experiences and has said that he wants viewers to hate the character for embodying gamer stereotypes. Boogie2988 videos range from absurd rants to serious discussions on daily life, such as the ethics of paid promotion on YouTube channels. He won the Trending Gamer award at The Game Awards 2016. 
 (4) Joseph Michael Graceffa (born May 16, 1991) is an American YouTube personality, actor, author, producer, and singer. He has two active YouTube channels, both of which are named after him. One of them is for vlogging and the other is used for video gaming content. They both have a combined total of more than 1.7 billion views. His other YouTube channel, which he no longer uses, called WinterSpringPro, has 60 million views (as of August 2017). He was a contestant on the 22nd and 24th seasons of "The Amazing Race" and has appeared in short films (as web films) and short web television series on YouTube and its subscription-only service YouTube Red. 
 Question: Shane and Friends have interviewed an American YouTube personality who has two active YouTube channels, which have a combined total of how many views? |
| Initial Prompt | Use the relevant information provided to answer the multi-hop reasoning question. Please provide the most accurate and concise answer to the question based on the information provided. The answer should be directly from the dataset, with no extra details or explanations. For yes-no questions, respond with either 'yes' or 'no.' Avoid long or complex sentences. |
| Learned Prompt | Use the relevant information provided to answer the multi-hop reasoning question. Identify the American YouTube personality mentioned and determine the combined total views of their two active YouTube channels. Provide the most accurate and concise answer directly from the dataset, without extra details or explanations. For yes-no questions, respond with either 'yes' or 'no.' Avoid long or complex sentences. Focus on extracting the specific information needed to answer the question accurately. Ensure the answer is clear and directly addresses the query. |

Table 6: Training cost of MetaTuner against prompt optimization and fine-tuning methods (hour).

| Method | MetaTuner | PPO | DPO | BPO | RLPrompt |
|---|---|---|---|---|---|
| Qwen2.5-7B | $\approx 5$ | $\approx 3.5$ | $\approx 3$ | $\approx 3.5$ | $\approx 2.5$ |
| Qwen2.5-3B | $\approx 3.5$ | $\approx 2.5$ | $\approx 2$ | $\approx 2$ | $\approx 1.5$ |

Table 7: Average inference cost of Qwen2.5-3B base model per query (second).

| Dataset | MATH | GSM8K | HotpotQA | CosmosQA | Avg Cost | Avg Best Performance |
|---|---|---|---|---|---|---|
| Answer Generation | 0.43 | 0.39 | 0.07 | 0.06 | 0.24 | 52.38 |
| +Prompt | 0.6(+0.17) | 0.55(+0.16) | 0.23(+0.16) | 0.18(+0.12) | 0.39(+0.15) | 57.57 |
| +Prompt & Parameter | 0.73(+0.30) | 0.66(+0.27) | 0.25(+0.18) | 0.48(+0.42) | 0.53(+0.29) | 65.10 |

Table 8: Average inference cost of Qwen2.5-7B base model per query (second).

| Dataset | MATH | GSM8K | HotpotQA | CosmosQA | Avg Cost | Avg Best Performance |
|---|---|---|---|---|---|---|
| Answer Generation | 0.71 | 0.68 | 0.14 | 0.11 | 0.41 | 60.30 |
| +Prompt | 0.99(+0.28) | 0.95(+0.27) | 0.44(+0.30) | 0.36(+0.25) | 0.69(+0.28) | 64.68 |
| +Prompt & Parameter | 1.48(+0.77) | 1.48(+0.80) | 0.47(+0.33) | 0.48(+0.37) | 0.98(+0.57) | 68.60 |

## C.4 EFFICIENCY ANALYSIS

**Training efficiency.** We take the MATH dataset as an example to compare the training overhead of MetaTuner's joint training process against conventional prompt optimization and fine-tuning methods. All training procedures are carried out on two 80GB GPUs, and the associated costs are reported in Table 6. Since MetaTuner collects training samples online, to ensure a fair comparison, we include the time required for additional data collection in the training cost of the baselines as well. From the results, MetaTuner indeed incurs higher training costs compared to baseline approaches. However, since this training is performed only once, we consider the training cost to be practically acceptable.

**Inference efficiency.** During the inference stage, for a given query $x$, MetaTuner needs to generate a set of prompts and parameters to complete the downstream task. Unlike the standard text generation task, there are additional costs associated with the LoRA parameter generation. In practice, we generate parameters for the o_proj part of the decoder layers in the downstream actor model $\mathcal{M}$. As mentioned in Section 3.4, each LoRA parameter is generated by a hyper-network $\phi_q = \{W_d^b \in \mathbb{R}^{d_\mathcal{M} \times l}, W_u^b \in \mathbb{R}^{h_\mathcal{G} \times r_\mathcal{M}}, W_d^a \in \mathbb{R}^{r_\mathcal{M} \times l}, W_u^a \in \mathbb{R}^{h_\mathcal{G} \times k_\mathcal{M}}\}$. Therefore, depending on the task scenario and model choice, the additional cost varies with sequence length $l$, the hidden size $h_\mathcal{G}$ of the prompt generator, the LoRA rank $r_\mathcal{M}$ of the generated parameters, and the number of decoder layers $K$ in the downstream actor model. However, compared to the parameter quantity of the LLM itself, the hyper-network is relatively small, so it does not incur significant computational overhead.

In Table 7 and Table 8, we report the cost of generating prompts and LoRA parameters, respectively. For reference, we also present the original inference cost without generating prompts and LoRA parameters, as well as the average best performance for each category of methods. From the above results, we can see: (1) Compared to methods with prompt generation, MetaTuner achieves an absolute performance gain of 5.73% at the cost of an average additional inference latency of 0.22 seconds. (2) Compared to Vanilla and Fine-tuning methods, MetaTuner achieves an absolute performance gain of 10.51% at the cost of an average additional inference latency of 0.43 seconds.

**Improving inference efficiency.** To accelerate the inference process, we further propose MetaTuner-C, which directly clusters the queries in the test set and then uses the prompt and parameters associated with the cluster centroid for all samples within that cluster during inference. Assuming the test set contains N samples and is partitioned into K clusters, the additional time overhead of MetaTuner-C compared to vanilla methods can be reduced to $t = K(t_p + t_q)/N$, where $t_p$ and $t_q$ denote the time costs for generating the prompt and parameters for a single query, respectively. Notably, our main method, MetaTuner-J, can be viewed as a special case of MetaTuner-C when $K = N$. We evaluate MetaTuner-C with various $K$ using the Qwen2.5-3B base model on the MATH dataset and report the results in Table 9 and Table 10. Our analysis shows that MetaTuner-C achieves strong performance while significantly reducing inference time. For example, under the Qwen2.5-3B setting, using the best fine-tuning baseline DPO as a reference, MetaTuner-C with K=50 attains an absolute performance gain of 3.41% with only 0.03 seconds of additional latency per query.

Table 9: Performance of MetaTuner-C with Qwen2.5-3B model on MATH.

| Method | MetaTuner-J | MetaTuner-C (K=50) | MetaTuner-C (K=30) | MetaTuner-C (K=10) | DPO |
|---|---|---|---|---|---|
| Performance | 41.33 | 40.00 | 39.11 | 37.56 | 36.59 |
| Inference Cost (s) | 0.73(+0.30) | 0.46(+0.03) | 0.45(+0.02) | 0.437(+0.007) | 0.43 |

Table 10: Performance of MetaTuner-C with Qwen2.5-7B model on MATH.

| Method | MetaTuner-J | MetaTuner-C (K=50) | MetaTuner-C (K=30) | MetaTuner-C (K=10) | DPO |
|---|---|---|---|---|---|
| Performance | 48.67 | 45.56 | 45.11 | 44.89 | 43.78 |
| Inference Cost (s) | 1.48(+0.77) | 0.80(+0.09) | 0.76(+0.05) | 0.73(+0.02) | 0.71 |

# D IMPLEMENTATION DETAILS

## D.1 PRELIMINARY EXPERIMENT

We present the prompts used in the right subfigure of the preliminary experiment below. For constructing the training data, the prompt $p$ is concatenated in front of the query $x$ to form the model input, while the ground truth $y$ serves as the expected output. The supervised fine-tuning process is conducted using Llama-Factory (Zheng et al., 2024), with the epoch set to 1, the batch size set to 64, and the learning rate tuned over the values $\{5e\text{-}4, 1e\text{-}4\}$.

---

**Prompts for Supervised Fine-Tuning on HotpotQA.**

## Prompt_0
From the given text, define or describe a particular term, concept, or idea mentioned. Your response should be based solely on how the term is presented in the passage.

## Prompt_1
Based on the passage, identify the character or person associated with the provided description. Provide the correct name or identifier, as supported by the text.

## Prompt_2
Use the passage to infer the implied meaning or consequence of a particular event, decision, or fact. Provide the most logical answer based on the information available.

## Prompt_3
Using the passage provided, identify the location or place associated with the subject matter. Make sure the answer is clearly supported by information found in the text.

## Prompt_4
Read the provided text and identify specific facts related to a particular subject. Answer the question using only information found in the passage.

## Prompt_5
Compare the details given in the passage and answer the question by identifying similarities or differences between two subjects or pieces of information.

## Prompt_6
From the given text, determine the timeline of events or specific dates related to a topic. Answer the question with precise chronological details from the passage.

## Prompt_7
Retrieve a specific detail or piece of information from the passage to answer a question. Make sure the answer directly aligns with what is stated in the text.

---

## Prompt_8
Given the passage, answer a comparative question by identifying similarities or differences between two concepts, characters, or events described in the text.

## Prompt_9
Identify the relationship between two subjects or concepts described in the passage. Answer the question by focusing on how these two entities are connected or related.

## Prompt_10
From the passage, answer the question by explaining the situation or problem presented in the text. Provide a clear response that addresses the core issue or situation described.

## Prompt_11
Read the passage and summarize the key points that answer the question. Focus on providing a concise, clear response that captures the main elements of the text.

## Prompt_12
Given a passage, answer the question by making an inference based on the information provided. Your answer should logically follow from the details in the text, even if it's not explicitly stated.

## Prompt_13
From the passage, answer the question by providing a summary of the main events or points described. Your answer should focus on the key takeaways and provide a concise overview of what is happening.

## Prompt_14
Based on the passage, explain a process or sequence of events described in the text. Provide the steps or order of occurrences in your response.

## Prompt_15
From the passage, answer the question by considering the broader context in which the information is provided. Your answer should reflect an understanding of the context and its relevance to the question.

## Prompt_16
Based on the passage, answer the question by extracting the most important details that would influence a decision or action. Your answer should be based on the most relevant information provided in the text.

## Prompt_17
From the provided passage, answer the question by identifying the key detail or piece of information that directly addresses the question.

## Prompt_18
Given a passage, extract the relevant information that answers a specific question. Ensure that your response is directly supported by details in the passage.

## Prompt_19
Based on the passage, answer the question by identifying the character or event described. Provide the answer that best fits the situation outlined in the text.

### D.2 IMPLEMENTATION OF METATUNER

**Data splition**. We split each dataset into training, validation, and test sets for model training and evaluation. For MATH, we use the 12K instances in the training set as specified in (Lightman et al.,

Table 11: Performance after the warm-up stage.

| Base Model | MATH | GSM8K | HotpotQA | CosmosQA |
|---|---|---|---|---|
| Qwen2.5-7B | 45.33 | 77.48 | 52.20 | 90.05 |
| Qwen2.5-3B | 35.33 | 71.27 | 53.95 | 84.00 |

Table 12: Performance of Qwen2.5-3B on MATH with smaller expert datasets.

| Data Quantity | 100% | 60% | 20% |
|---|---|---|---|
| Performance After Warm-up | 35.33 | 36.67 | 34.44 |
| Performance After Joint Training | 41.33 | 41.33 | 40.89 |

2024), and randomly divide the remaining 500 instances into 50 for validation and 450 for testing. For GSM8K, the official test set contains 1,319 instances, so we allocate 473 instances for validation and the remaining 7K instances for training. For HotpotQA, we sample 20,000 instances from the official training set and randomly select 500 for validation and 2,000 for testing from the official dev-set-distractor. For CosmosQA, we use the entire 25,262 instances in the training set and randomly sample 500 and 2,000 instances from the test set for validation and testing, respectively.

**Details of the warm-up stage**. For the warm-up of $\mathcal{M}$, we use the queries $x$ from the original dataset as model inputs and the corresponding ground truth $y$ as model outputs for supervised fine-tuning. For the warm-up of $\mathcal{G}$, since there is no ground truth prompt, we construct the supervised training data with rejection sampling. Specifically, we first utilize a leading closed-source LLM to generate expert prompts $\mathcal{D}_{\text{expert}} = \{\hat{p}_i\}_{i=1}^N$, and then use the task reward $\mathcal{R}$ to filter out positive examples. The positive samples are then collected as $\mathcal{D}_{\text{PO}} = \{(x_i, \hat{p}_i) | \mathcal{R}(\mathcal{M}^*(\hat{p}_i, x_i)) = 1\}_{i=1}^N$ for the supervised fine-tuning of prompt generation. Here, $\mathcal{M}^*$ and $\mathcal{G}^*$ represent the model after warm-up. Naturally, the pre-warmed model can form a pipeline $\hat{y}_i = \mathcal{M}^*(\mathcal{G}^*(\tilde{p}, x_i), x_i)$ to complete downstream tasks. We report the results in Table 11.

To further investigate the impact of generated expert data during the warm-up stage, we additionally study how using a smaller expert dataset (in Table 12) or a weaker expert model (in Table 13) affects the MetaTuner's final performance. Our analysis reveals that MetaTuner exhibits robust performance under both weaker and smaller expert dataset settings. As the size of the expert dataset decreases, performance remains stable both after warm-up and joint training. When a weaker expert model is used for warm-up, the post-warm-up performance degrades moderately as the expert model's capability diminishes. However, this performance gap is largely mitigated after joint training. This suggests that the warm-up phase only requires a basic level of prompt-generation competence. The proposed MetaTuner framework can subsequently discover superior prompt–parameter combinations during joint training phase, thereby enhancing overall performance.

**Construction of the meta prompts**. In this paper, since we adopt a prompt-rewrite paradigm for prompt generation, each meta prompt contains a query and an initial prompt, as well as an instruction to describe the prompt rewrite task. We follow the guidance of (Yang et al., 2024b) to construct a base version of the prompt for each task as the initial prompt, which only includes a description of the task and the required answer format to facilitate answer parsing. For GSM8K, since the base model already produces highly consistent answer formats, we omit the answer format requirement in the initial prompt. The meta prompts on four datasets are presented below.

Table 13: Performance of Qwen2.5-3B on MATH with weaker expert datasets.

| Expert Model | Proprietary LLM | Qwen2.5-32B | Qwen2.5-14B |
|---|---|---|---|
| Performance After Warm-up | 35.33 | 33.11 | 31.33 |
| Performance After Joint Training | 41.33 | 40.22 | 40.00 |

---

**Meta prompt for MATH dataset.**

Given a query and an initial prompt, please add more detailed guidance to the initial prompt to help the model better respond to the question. The generated prompt will be concatenated in front of the query to assist the model in answering questions. During the rewriting process, ensure that the new prompt includes the information from the original prompt, such as the task description and the answer format requirements. The generated new prompt should not exceed 100 words.

Query:
{query}

Initial Prompt:
Given a complex mathematical problem, solve it step by step using a chain-of-thought approach. The answer and reasoning process should not exceed 250 words. Present the final result in LaTeX using a "\boxed{}" without any units.

New Prompt:

---

**Meta prompt for GSM8K dataset.**

Given a query and an initial prompt, please add more detailed guidance to the initial prompt to help the model better respond to the question. The generated prompt will be concatenated in front of the query to assist the model in answering questions. During the rewriting process, ensure that the new prompt includes the information from the original prompt, such as the task description and the answer format requirements. The generated new prompt should not exceed 100 words.

Query:
{query}

Initial Prompt:
Given a complex mathematical problem, solve it step by step using a chain-of-thought approach.

New Prompt:

---

**Meta prompt for HotpotQA dataset.**

Given a query and an initial prompt, please add more detailed guidance to the initial prompt to help the model better respond to the question. The generated prompt will be concatenated in front of the query to assist the model in answering questions. During the rewriting process, ensure that the new prompt includes the information from the original prompt, such as the task description and the answer format requirements. The generated new prompt should not exceed 100 words.

Query:
{query}

Initial Prompt:
Use the relevant information provided to answer the multi-hop reasoning question. Please provide the most accurate and concise answer to the question based on the information provided. The answer should be directly from the dataset, with no extra details or explanations. For yes-no questions, respond with either 'yes' or 'no.' Avoid long or complex sentences.

New Prompt:

---

**Meta prompt for CosmosQA dataset.**

Given a query and an initial prompt, please add more detailed guidance to the initial prompt to help the model better respond to the question. The generated prompt will be concatenated in front of the query to assist the model in answering questions. During the rewriting process, ensure that the new prompt includes the information from the original prompt, such as the task description and the answer format requirements. The generated new prompt should not exceed 100 words.

Query:
{query}

Initial Prompt:
Based on the provided context, select the most relevant and accurate option as the correct answer to the question. The answer should be directly related to the context, and must focus on the key details or events described.

New Prompt:

---

