# OpenReview forum: "Prompt and Parameter Co-Optimization for Large Language Models"
_ICLR.cc/2026/Conference — ICLR 2026 Poster_

### Official Review · Reviewer_LrBX · 2025-10-29

**Soundness:** 2
**Presentation:** 3
**Contribution:** 2
**Rating:** 2
**Confidence:** 3

**Summary:**

This paper proposes MetaTunner which is a framework that jointly optimizes both prompts and model parameters for finetuning LLMs. They design a prompt generator that outputs discrete prompts and a parameter generator that produces LoRA style updates, this two modules are trained jointly, with an additional ad-hoc supervised regularization loss on prompt. Experimetns on various QA datasets such as MATH, GSM8K show their proposed method outperforms various baselines including prompt tuning, LoRA tuning, and the closely related hybrid BetterTogether approach.

**Strengths:**

1. This paper address an important direction that the prompt for a downstream tasks need to be engineereed and might not be optimial, finetuning both prompt and model offers a more unified task adaptation framework.
2. The propose architecture is clear and experiment results are good.

**Weaknesses:**

1. The proposed framework is essentially a composition of existing components: prompt generation + LoRA + shared encoder, which is incremental and lack clear conceptual novelty. The idea of optimizing both prompt and model itself is well-explored in studies like BetterTogether, which the authors themselves cite but weakly differentiate from.
2. Instead using soft prompts that can truely trained jointly end-to-end, it is questionable why this paper choose discrete prompts generated by a separate LLM, although it may help interpretability. More insights or analysis on this should be discussed for why discrete prompts are preferable.
3. The supervised regularization loss to handle discrete-continuous issues feels ad hoc and unstable, it is based on an auxiliary “expert prompt” dataset, and the algorithm itself has no analysis of convergence.
4. The evaluation is limited to 4 QA benchmarks and a single Qwen model family with two different sizes, the method's generalizability to other domains and models remains untested.

**Questions:**

See weaknesses.

---

> ### Author Response · Authors · 2025-11-22
> **Rebuttal by Authors [W1, W2]**
>
> **[W1] The proposed framework is essentially a composition of existing components: prompt generation +** **LoRA** **+ shared encoder, which is incremental and lack clear conceptual novelty. The idea of optimizing both prompt and model itself is well-explored in studies like BetterTogether, which the authors themselves cite but weakly differentiate from.**
>
> Thanks for the comments.
>
> - Although our method builds upon existing techniques (e.g., prompt optimization and fine-tuning), **our contribution is not merely a straightforward combination of off-the-shelf components. Instead,**
>
>   - We propose **a unified end-to-end framework** that jointly optimizes discrete prompts and continuous LoRA parameters, with a **parameter sharing mechanism** to enable cross-task knowledge transfer.
>   - We introduce **a regularized supervision loss** to guide the joint learning process, stabilizing the training process and improving the generalization capability.
>   -  To the best of our knowledge, this work presents **the first end-to-end framework** for joint training of prompts and model parameters.
>
> - In contrast to staged approaches such as BetterTogether, **MetaTuner updates the prompt generator and LoRA generator simultaneously within the same training process, under a shared representation**, making them interdependent rather than sequentially chained. Results across four datasets demonstrate that our parameter-sharing architecture combined with the regularized supervision loss enables effective exploration in the joint prompt–parameter space, leading to superior empirical performance.
>
>
>
> **[W2] Instead using soft prompts that can truely trained jointly end-to-end, it is questionable why this paper choose discrete prompts generated by a separate LLM, although it may help interpretability. More insights or analysis on this should be discussed for why discrete prompts are preferable.**
>
> We sinscerely thank the reviwer for the insightful comments.
>
> Beyond improved interpretability, we believe discrete prompts offer several additional advantages over soft prompts:
>
> 1. **Training robustness**: Compared to soft prompts, the optimization of discrete prompts is more stable. Existing work [1,2] has shown that soft prompt tuning can suffer from “representation collapse,” where prompts for different inputs converge to nearly identical vectors. In MetaTuner, we optimize discrete prompts, i.e., prompts must take the form of readable natural language, with each token subject to clear semantic constraints. This forces the optimization process to search over structured, human-interpretable instructions rather than arbitrary continuous vectors, thereby reducing the risk of overfitting. To further validate this, we conduct a comparison experiment, and the results show that discrete prompts achieve better performance in co-optimization settings.
>
>    - Performance comparison between MetaTuner and Soft Co-optimization with Qwen2.5-3B across four datasets.
>
>    | Method   | MetaTuner | Soft Co-optimization |
>    | -------- | --------- | -------------------- |
>    | MATH     | 41.33     | 26.22                |
>    | GSM8K    | 73.08     | 53.30                |
>    | HotpotQA | 58.85     | 49.35                |
>    | CosmosQA | 87.15     | 50.65                |
>
>    - Performance comparison between MetaTuner and Soft Co-optimization with Qwen2.5-7B across four datasets.
>
>    | Method   | MetaTuner | Soft Co-optimization |
>    | -------- | --------- | -------------------- |
>    | MATH     | 48.67     | 38.67                |
>    | GSM8K    | 78.92     | 62.24                |
>    | HotpotQA | 54.56     | 43.65                |
>    | CosmosQA | 92.25     | 82.80                |
>
> 2. **Transferability**: Discrete prompts also exhibit superior transferability compared to soft prompts. As highlighted by Reviewer RbJU, a key advantage of prompt-based methods is infrastructure simplicity—enabling a single frozen model to serve multiple tasks efficiently. In contrast, soft prompts are implemented by inserting learnable parameters at specific positions in the input sequence, and their dimensionality and length are tightly coupled to the architecture of the backbone model. Consequently, soft prompts trained on one model are generally not transferable to others. In contrast, discrete prompts are represented as natural language, making them inherently portable across different models without architectural adaptation.
>
> [1] Vector-Quantized Input-Contextualized Soft Prompts for Natural Language Understanding
>
> [2] Representation Projection Invariance Mitigates Representation Collapse

---

> ### Author Response · Authors · 2025-11-22
> **Rebuttal by Authors [W3]**
>
> **[W3] The supervised** **regularization** **loss to handle discrete-continuous issues feels ad hoc and unstable, it is based on an auxiliary “expert prompt” dataset, and the algorithm itself has no analysis of convergence**.
>
> We sincerely thank the reviewer for this valuable feedback. To address this concern, we first provide a detailed analysis on the influence of the expert prompt dataset, and then provide empirical evidence of the training dynamics and convergence behavior of MetaTuner.
>
> 1. **Expert prompt analysis.**
>
> The training of MetaTuner consists of two phases: the warm-up phase and the joint training phase.
>
> - During the warm-up phase, we pre-train the prompt generator using expert data generated by GPT-4o, enabling the model to develop strong instruction-following capabilities for prompt generation.
> - In the joint training phase, we find that collecting expert data using the policy being trained itself—rather than relying on GPT-4o—yields better results. A detailed analysis of this observation is provided in Section 4.4.
>
> To further investigate the impact of GPT-4o–generated expert data during the warm-up stage, we additionally study how using a **weaker expert model** or a **smaller expert dataset** affects MetaTuner’s final performance. Below are the experimental results on the MATH dataset using the Qwen2.5-3B base model.
>
> - Performance of Qwen2.5-3B on MATH with smaller expert datasets.
>
> | Data Quatity                     | 100%  | 60%   | 20%   |
> | -------------------------------- | ----- | ----- | ----- |
> | Performance After Warmup         | 35.33 | 36.67 | 34.44 |
> | Performance After Joint Training | 41.33 | 41.33 | 40.89 |
>
> - Performance of Qwen2.5-3B on MATH with weaker expert datasets.
>
> | Expert Model                     | GPT-4o | Qwen2.5-32B | Qwen2.5-14B |
> | -------------------------------- | ------ | ----------- | ----------- |
> | Performance After Warmup         | 35.33  | 33.11       | 31.33       |
> | Performance After Joint Training | 41.33  | 40.22       | 40.00       |
>
> Our analysis reveals that MetaTuner exhibits robust performance under both weaker and smaller expert dataset settings:
>
> - As the size of the expert dataset decreases, performance remains stable both after warm-up and after joint training.
> - When a weaker expert model is used for warm-up, the post-warm-up performance degrades moderately as the expert model’s capability diminishes. However, this performance gap is largely mitigated after joint training.
>
> This suggests that the warm-up phase only requires a basic level of prompt-generation competence. The proposed MetaTuner framework can subsequently discover superior prompt–parameter combinations during joint training, thereby enhancing overall performance.
>
> 2. **Convergence analysis.**
>
> Here, we report the training loss curves of the Qwen2.5-3B and Qwen2.5-7B on the MATH dataset as empirical validation of the convergence behavior of the proposed method. The loss curve can be found via https://drive.google.com/file/d/16h6LEr0DhbiJPGxlsXR1RgE0fLjltMDm/view?usp=sharing.
>
> Our analysis shows that the loss exhibits a consistently stable and steady downward trend. The rapid initial drop indicates that the model quickly captures effective patterns in the data, followed by a gradual slowdown in the rate of decrease, eventually stabilizing at a low level. This demonstrates that the training process converges reliably and stably.

---

> ### Author Response · Authors · 2025-11-22
> **Rebuttal by Authors [W4]**
>
> **[W4] The evaluation is limited to 4 QA benchmarks and a single Qwen model family with two different sizes, the method's generalizability to other domains and models remains untested.**
>
> We sincerely thank the reviewer for this constructive suggestion regarding the scope of our evaluation. To further validate the effectiveness of MetaTuner across broader scenarios and model architectures, we additionally conduct experiments on the chess-playing task (Checkmate-in-One-Move dataset) using the Llama3.2-3B-Instruct base model. The results show that **MetaTuner consistently outperforms baseline methods**, demonstrating the efficacy of our proposed prompt–parameter co-optimization framework, which is built upon parameter sharing and a supervised regularization loss.
>
> | Method         | Qwen2.5-3B on Checkmate in One Move | Llama3.2-3B-Instruct on MATH |
> | -------------- | ----------------------------------- | ---------------------------- |
> | Vanilla        | 0.00                                   | 21.33                        |
> | GPT-4o         | 29.71                               | 18.89                        |
> | RLPrompt       | 0.00                                   | 24.00                        |
> | BPO            | 0.00                                   | 26.22                        |
> | OPRO           | 0.00                                   | 25.77                        |
> | CFPO           | 0.00                                   | 25.33                        |
> | SFT            | 13.71                               | 24.44                        |
> | PPO            | 14.00                               | 23.78                        |
> | DPO            | 13.42                               | 26.44                        |
> | KTO            | 14.57                               | 29.78                        |
> | BetterTogether | 15.14                               | 30.44                        |
> | MetaTuner-J    | 17.71                               | 31.78                        |

---

> > ### Comment · Reviewer_LrBX · 2025-11-25
> >
> > Thanks for providing additional results to justify the proposed method. Despite the empirical results, I am not fully convinced by the novelty or the elegance of the method. I will increase the score to 4 given the effort on the experiments.

---

> ### Author Response · Authors · 2025-11-30
> **Response by Authors**
>
> Thank you very much for your thoughtful feedback. We sincerely appreciate your time and constructive engagement with our work.
>
> To further clarify the novelty and elegance of MetaTuner, we summarize key distinctions from prior joint training approaches (e.g., BetterTogether) in the table below. Unlike staged co-optimization methods, MetaTuner introduces a unified end-to-end framework that (1) jointly optimizes discrete prompts and continuous LoRA parameters via a shared meta encoder, (2) employs a regularized supervision loss to stabilize co-optimization, and (3) enables online exploration of effective prompt–parameter combinations, and (4) results in tighter coupling, better generalization, and consistent gains.
>
> | Aspect                 | BetterTogether                                               | MetaTuner                                                    | Advantage of MetaTuner                                       |
> | ---------------------- | ------------------------------------------------------------ | ------------------------------------------------------------ | ------------------------------------------------------------ |
> | Training Paradigm      | Staged optimization: prompts and parameters are updated alternately in separate phases. | End-to-end joint training: prompt and LoRA parameters are optimized simultaneously in a single loop. | (1) Enables genuine co-adaptation between prompts and parameters; (2) Eliminates error propagation inherent in staged updates. |
> | Model Structure        | Independent prompt generator and LoRA adapters, with no explicit parameter sharing. | Shared meta encoder to generate prompts and LoRA weights via a unified representation. | (1) Fosters synergistic interaction through a unified generative process; (2) Enhances cross-task generalization via shared representations. |
> | Optimization Objective | Typically uses standard task loss, e.g. SFT Loss for both prompt and parameter optimization | Introduces a regularized supervision loss that couples prompt and LoRA learning, stabilizing joint optimization. | (1) Improves training stability; (2) Enables online exploration of effective prompt–parameter combinations during training. |
> | Empirical Performance  | Strong baseline, but performance plateaus on diverse-task settings. | Consistently outperforms BetterTogether across 4 datasets (+6.62% absolute improvement on average) | (1) Achieves consistent empirical gains across diverse tasks; (2) Validates the effectiveness of our joint architecture and optimization design. |
>
> Besides, in response to your earlier comments, we have further expanded our experimental scope and provide additional results below.
>
> - We further **validate the advantage of MetaTuner against soft co-optimization method across** **multiple base models** (including LLaMA3.2-3B,  Llama3.1-8B, and Mistral-7B).
>
> | Base Model| MetaTuner | Soft Co-optimization |
> | --------------------- | --------- | -------------------- |
> | Qwen2.5-3B | 41.33| 26.22|
> | Qwen2.5-7B| 48.67| 38.67|
> | Llama-3.2-3B-Instruct | 31.78| 26.44|
> | Llama-3.1-8B-Instruct | 35.78| 30.67|
> | Mistral-7B-Instruct| 18.44| 14.44 |
>
> - We further **expand evaluation to more datasets and diverse model settings**, where MetaTuner shows stable performance improvements.
>
> | Method | Qwen2.5-3B on Checkmate in One Move | Qwen2.5-7B on Checkmate in One Move | Llama3.2-3B-Instruct on Checkmate in One Move | Llama3.2-3B-Instruct on MATH | Mistral-7B-Instruct on MATH | Llama3.1-8B-Instruct on MATH |
> | -------------- | ----------------------------------- | ----------------------------------- | --------------------------------------------- | ---------------------------- | --------------------------- | ---------------------------- |
> | Vanilla| 0.00 | 0.00| 0.00| 21.33| 10.89| 25.11|
> | GPT-4o| 29.71| 29.71| 29.71| 18.89| 18.89| 18.89|
> | RLPrompt| 0.00| 0.00| 0.00| 24.00| 12.00| 25.56|
> | BPO| 0.00| 0.86| 1.43| 26.22| 12.44| 26.22|
> | OPRO| 0.00| 0.57| 1.14| 25.77| 11.78| 27.33|
> | CFPO| 0.00| 0.00| 1.43| 25.33| 11.56| 27.11|
> | SFT| 13.71| 18.00| 18.85| 24.44| 14.00| 33.11|
> | PPO| 14.00| 18.29| 19.43| 23.78| 14.67| 32.22|
> | DPO| 13.42| 18.86| 19.14| 26.44| 16.00| 34.00|
> | KTO| 14.57| 18.57| 20.00| 29.78| 13.78| 33.56|
> | BetterTogether| 15.14| 19.71|20.57| 30.44| 16.44| 34.44|
> | MetaTuner-J| 17.71| 20.86| 21.43| 31.78| 18.44| 35.78|
>
> We hope these clarifications and results help illustrate both the conceptual distinction and practical effectiveness of our approach. Thank you again for your valuable feedback.

---

### Official Review · Reviewer_RbJU · 2025-10-31

**Soundness:** 3
**Presentation:** 3
**Contribution:** 3
**Rating:** 8
**Confidence:** 3

**Summary:**

The paper proposes a joint framework for LLM prompt optimization and parameter training. Clear gains are demonstrated over prompt optimization and fine-tuning.

**Strengths:**

- Clear and concise description of the approach
- The evaluation covers a very good range of prompt optimization and fine-tuning methods
- Strong and consistent gains over the baselines across 4 datasets and 2 model sizes
- Interesting and well-designed ablation studies

**Weaknesses:**

- A major benefit of prompt optimization is infrastructure simplicity and efficiency - serving the same frozen model for multiple tasks is much cheaper than multiple dedicated models. This benefit is lost here.
- The whole system design is quite complex, but the strong gains and comprehensive ablations make up for it
- Given the otherwise broad experimental setup, I find it a bit unfortunate that - iiuc - fine-tuning is limited to LoRA. An isolated small-scale experiment with full fine-tuning of the 3B model would tell us more about the head room and would help to better understand the gains of MetaTuner.

Minor comments:
- L107: "In LLMs, we typically work with input-output pairs" is too simplistic/general - e.g. think LLM pre-training etc.
- Fig. 1 font size is too tiny

**Questions:**

- Can you say something about the computational training costs compared to regular prompt optimization / fine-tuning?

---

> ### Author Response · Authors · 2025-11-22
> **Rebuttal by Authors**
>
> **[W1] A major benefit of prompt** **optimization** **is infrastructure simplicity and efficiency - serving the same frozen model for multiple tasks is much cheaper than multiple dedicated models. This benefit is lost here.**
>
> We sincerely thank the reviewer for the thoughtful observation regarding infrastructure simplicity and deployment efficiency.
>
> 1. Indeed, MetaTuner does sacrifice some degree of deployment simplicity compared to pure prompt optimization methods. However, we view this as **an intentional trade-off between generality and specialization**: by allowing limited, efficient parameter adaptation alongside prompt generation, MetaTuner achieves substantially stronger performance.
> 2. However, it is important to emphasize that only lightweight LoRA weights are generated per-query, while the backbone LLM remains entirely frozen. At inference time, **the generated parameters can be merged on-the-fly or cached, following the standard “frozen model + LoRA” deployment** **paradigm**, which is far more lightweight than maintaining fully fine-tuned models.
> 3. Moreover, MetaTuner is structurally flexible: **when parameter adaptation is unnecessary, one can disable the** **LoRA** **head and use only the prompt generator (referred to as MetaTuner-P)**. Notably, MetaTuner-P still outperforms conventional prompt optimization baselines (see the table below ), demonstrating the superiority of its learned prompting mechanism even without parameter adaptation.
>
> | Base Model | MetaTuner-P | RLPrompt | BPO   | OPRO  | CFPO  |
> | ---------- | ----------- | -------- | ----- | ----- | ----- |
> | Qwen2.5-3B | 26.89       | 22.67    | 26.44 | 21.00 | 17.78 |
> | Qwen2.5-7B | 34.89       | 31.33    | 32.67 | 22.00 | 27.11 |
>
>
>
> **[W2 & Q1] The whole system design is quite complex, but the strong gains and comprehensive ablations make up for it. Can you say something about the computational training costs compared to regular prompt** **optimization** **/ fine-tuning?**
>
> Thanks for the comments. We take the MATH dataset as an example to compare the training overhead of MetaTuner’s joint training process against conventional Prompt Optimization and Fine-tuning methods. Since MetaTuner collects training samples online, to ensure a fair comparison, we include the time required for additional data collection in the training cost of the baselines as well. From the results, MetaTuner indeed incurs higher training costs compared to baseline approaches. However, since this training is performed only once, we consider **the training cost to be practically acceptable.**
>
> | Method     | MetaTuner | PPO   | DPO  | BPO   | RLPrompt |
> | ---------- | --------- | ----- | ---- | ----- | -------- |
> | Qwen2.5-7B | ~5h       | ~3.5h | ~3h  | ~3.5h | ~2.5h    |
> | Qwen2.5-3B | ~3.5h     | ~2.5h | ~2h  | ~2h   | ~1.5h    |
>
>
>
> **[W3] Given the otherwise broad experimental setup, I find it a bit unfortunate that - iiuc - fine-tuning is limited to LoRA. An isolated small-scale experiment with full fine-tuning of the 3B model would tell us more about the head room and would help to better understand the gains of MetaTuner.**
>
> Thanks for the insightful comments. We perform full fine-tuning on the Qwen2.5-3B model on the MATH dataset, and compare it with LoRA-based SFT and MetaTuner’s LoRA-based regularized SFT. Due to practical constraints, we fine-tune all model parameters in the SFT(Full) method, while only generate full parameters of the o_proj layers in MetaTuner (Full).
>
> | Method      | MetaTuner(LoRA) | MetaTuner(Full) | SFT (LoRA) | SFT (Full) |
> | ----------- | --------------- | --------------- | ---------- | ---------- |
> | Performance | 41.33           | 41.78           | 29.33      | 30.44      |
>
> From the results, we can find that:
>
> 1. Under the current data scale and task setting, **switching from** **LoRA** **to full-parameter fine-tuning yields only marginal performance gains**, indicating limited headroom from unfreezing more parameters.
> 2. **MetaTuner’s gains are architectural, not parametric.** Even MetaTuner with full fine-tuning (0.4178) shows only a minor improvement over its LoRA variant (0.4133), while both dramatically outperform their SFT counterparts. This indicates that the core advantage of MetaTuner lies not in parameter count or update scope, but in its joint prompt–parameter co-optimization framework.
>
> **[Minor comments] (1) L107: "In LLMs, we typically work with input-output pairs" is too simplistic/general - e.g. think LLM pre-training etc. (2) Fig. 1 font size is too tiny**
>
> **Response:** We sincerely thank the reviewer for pointing out the readability issue. We will increase the font size in Fig. 1 and improve the overall clarity in the final version.

---

### Official Review · Reviewer_ttpV · 2025-11-01

**Soundness:** 2
**Presentation:** 2
**Contribution:** 2
**Rating:** 4
**Confidence:** 3

**Summary:**

In this work, a framework to jointly optimize prompt and parameter is proposed. The framework is named after metaTuner. During training time, the initial prompt is given to the meta encoder to rewrite with a language model A. In the meantime, there is a language model B to generate the parameter which is used to optimize the downstream actor model. The actor model takes the generated parameter and the the rewritten prompt as input to generate the answer. The response is used to compare with the ground truth and get the task specific reward, which is used to update the model A and B.

**Strengths:**

- The proposed approach is easy to follow and understand.
- The proposed approach has been verified in multiple SFT datasets.

**Weaknesses:**

- Concerns about the generalization of the model trained with the proposed approach. The model is trained to optimize for several specific task verticals, including prompt rewrite and parameter generation. one concern is about catastrophic forgetting: is the model still able to generalize to other unseen dataset during finetuning? Some OOD evaluation will helps.
- One hypothesis of the performance win is that the introduction of metaTuner actually increased the total number of parameters of the whole system and the scaling up of the model itself is the source of improvement. Is it possible to validate this point so we can confirm that the wins are from the approach instead of the model scaling up?

**Questions:**

- Concerns about the generalization of the model trained with the proposed approach. The model is trained to optimize for several specific task verticals, including prompt rewrite and parameter generation. one concern is about catastrophic forgetting: is the model still able to generalize to other unseen dataset during finetuning? Some OOD evaluation will helps.
- One hypothesis of the performance win is that the introduction of metaTuner actually increased the total number of parameters of the whole system and the scaling up of the model itself is the source of improvement. Is it possible to validate this point so we can confirm that the wins are from the approach instead of the model scaling up?

---

> ### Author Response · Authors · 2025-11-22
> **Rebuttal by Authors [W1 & Q1]**
>
> **[W1 & Q1] Concerns about the generalization of the model trained with the proposed approach. The model is trained to optimize for several specific task verticals, including prompt** **rewrite** **and parameter generation. one concern is about catastrophic forgetting: is the model still able to generalize to other unseen dataset during finetuning? Some OOD evaluation will helps.**
>
> We sincerely thank the reviewer for raising this important concern about generalization and potential catastrophic forgetting issue in MetaTuner.
>
> **First, we conduct an OOD experiment in Section 4.6 to explore the generalization capability of MetaTuner.** Specifically, we train the model on MATH, HotpotQA, and CosmosQA, and evaluate it on the GSM8K dataset. The results show that **MetaTuner achieves better out-of-distribution generalization compared to baseline methods.** This improvement primarily stems from our parameter-sharing mechanism. In MetaTuner, we formulate prompt generation and parameter generation as two distinct tasks that share a common meta encoder, which can be reframed as a multi-task learning (MTL) problem. Prior MTL theories [1,2,3,4,5,6] show that **such shared representations constrain the hypothesis space and promote learning of task-invariant features, thereby improving generalization**, which aligns well with our empirical results.
>
> **Second, we conduct additional OOD experiments on two model sizes under the other three additional dataset settings,** with the results shown below:
>
> - Train on MATH, HotpotQA, CosmosQA, Test on GSM8K
>
> | Method         | Qwen2.5-3B | Qwen2.5-7B |
> | -------------- | ---------- | ---------- |
> | MetaTuner      | 37.56      | 40.89      |
> | BetterTogether | 31.33      | 37.78      |
> | DPO            | 18.44      | 18.89      |
> | BPO            | 24.00      | 27.33      |
>
> - Train on MATH, GSM8K, CosmosQA, Test on HotpotQA
>
> | Method         | Qwen2.5-3B | Qwen2.5-7B |
> | -------------- | ---------- | ---------- |
> | MetaTuner      | 34.60      | 40.55      |
> | BetterTogether | 29.50      | 40.65      |
> | DPO            | 31.15      | 35.35      |
> | BPO            | 30.35      | 39.40      |
>
> - Train on MATH, GSM8K, HotpotQA, Test on CosmosQA
>
> | Method         | Qwen2.5-3B | Qwen2.5-7B |
> | -------------- | ---------- | ---------- |
> | MetaTuner      | 75.65      | 81.45      |
> | BetterTogether | 75.50      | 81.35      |
> | DPO            | 62.25      | 71.20      |
> | BPO            | 72.75      | 70.20      |
>
> From the results, we can observe consistent performance improvements of our framework compared with the other models, which demonstrate the generalization capability of our method.
>
> [1] Regularizing Neural Networks via Stochastic Branch Layers
>
> [2] Can Small Heads Help? Understanding and Improving Multi-Task Generalization
>
> [3] A Model of Inductive Bias Learning
>
> [4] The Benefit of Multitask Representation Learning
>
> [5] SHARING KNOWLEDGE IN MULTI-TASK DEEP REINFORCEMENT LEARNING
>
> [6] Multitask Learning via Shared Features: Algorithms and Hardness

---

> ### Author Response · Authors · 2025-11-22
> **Rebuttal by Authors [W2 & Q2]**
>
> **[W2 & Q2] One hypothesis of the performance win is that the introduction of MetaTuner actually increased the total number of parameters of the whole system and the scaling up of the model itself is the source of improvement. Is it possible to validate this point so we can confirm that the wins are from the approach instead of the model scaling up?**
>
> We sincerely thank the reviewer for the insightful question.
>
> Compared to pure Prompt Optimization or standard fine-tuning methods, MetaTuner does introduce additional parameterized modules. However, we argue that **beyond merely increasing the number of parameters, how parameters are structured and utilized is far more critical.** We validate this claim through the following experiments using the Qwen2.5-3B base model:
>
> 1. **We compare MetaTuner** (~6B total parameters) **against significantly larger frozen models** (GPT-4o, GPT-3.5, Qwen2.5-7B, Qwen2.5-14B) **as well as two supervised fine-tuned variants** (Qwen2.5-7B-SFT and Qwen2.5-14B-SFT). Despite having fewer parameters, MetaTuner achieves superior task performance, demonstrating its more efficient and effective use of parameters.
> 2. **We compare MetaTuner with a variant, MetaTuner (w/o S)**, which introduces even more parameters after remove the shared-parameter mechanism. Nevertheless, MetaTuner (w/o S) underperforms the original MetaTuner. This highlights that parameter organization matters more than sheer parameter count. In MetaTuner, parameter sharing not only reduces the total number of parameters but also enhances final task performance.
>
> | Model           | MATH  | GSM8K | HotpotQA | CosmosQA |
> | --------------- | ----- | ----- | -------- | -------- |
> | MetaTuner       | 41.33 | 73.08 | 58.85    | 87.15    |
> | GPT-4o          | 18.89 | 68.46 | 54.15    | 76.50    |
> | GPT-3.5         | 30.89 | 71.64 | 49.10    | 77.85    |
> | Qwen2.5-7B      | 18.44 | 51.63 | 19.85    | 36.80    |
> | Qwen2.5-14B     | 14.89 | 52.99 | 34.90    | 78.75    |
> | Qwen2.5-7B-SFT  | 41.33 | 61.41 | 43.20    | 82.65    |
> | Qwen2.5-14B-SFT | 26.22 | 66.41 | 40.50    | 84.35    |
> | MetaTuner w/o S | 40.22 | 72.10 | 57.55    | 86.65    |

---

> > ### Comment · Reviewer_ttpV · 2025-11-23
> > **Thanks for the response**
> >
> > Thanks for the response with additional results that decoupled the # of param and the shared-parameter mechanism. So looks like there are 1% wins compared with the "MetaTuner w/o S". Would be good to add these results to the paper for a more complete story. I have adjusted my rating accordingly.

---

> > > ### Author Response · Authors · 2025-11-24
> > > **Official Comment by Authors**
> > >
> > > Thanks very much for your thoughtful feedback. We greatly appreciate your constructive comments and the time you’ve taken to review our work. We will incorporate these results into the final version to further strengthen the work.

---

### Official Review · Reviewer_6Sni · 2025-11-01

**Soundness:** 4
**Presentation:** 3
**Contribution:** 4
**Rating:** 8
**Confidence:** 4

**Summary:**

This paper introduces MetaTuner, a unified framework for the joint optimization of prompts and model parameters in large language models (LLMs). Traditional approaches typically treat prompt optimization, discrete and external, and fine-tuning, continuous and internal, as separate processes. MetaTuner bridges these two paradigms through a shared meta-encoder that feeds into two complementary decoders: a prompt decoder that generates discrete natural language prompts and a parameter decoder that produces continuous LoRA-based parameter updates. To address the challenge of non-differentiability in prompt generation, the framework incorporates a supervised regularization loss that leverages an “expert dataset” of prompts, partly generated using GPT-4o. This mechanism stabilizes training and allows efficient co-optimization of both prompt and parameter spaces. Experiments on MATH, GSM8K, HotpotQA, and CosmosQA, using Qwen2.5-3B and Qwen2.5-7B backbones, demonstrate that MetaTuner achieves consistent performance gains over prompt-only, fine-tuning, and hybrid baselines such as BetterTogether.

**Strengths:**

Unified Framework: Introduces MetaTuner, a single architecture that jointly optimizes prompts and model parameters, integrating discrete prompt learning and continuous fine-tuning.

Dual-Decoder Design: Proposes a shared meta-encoder with separate prompt and parameter decoders that generate textual prompts and LoRA-based parameter updates in a coordinated manner.

Supervised Regularization: Employs a regularization loss guided by an expert prompt dataset to stabilize training and address non-differentiability in prompt optimization.

Proper Evaluation: Demonstrates consistent performance gains across multiple reasoning and QA benchmarks, supported by detailed ablations and generalization analyses.

In addition, the paper is well-written and proper examples are provided.

**Weaknesses:**

The paper could more clearly differentiate MetaTuner from sequential prompt–fine-tuning pipelines such as Chain-of-Thought distillation (Finlayson et al., 2025; Wei et al., 2022). A stronger conceptual contrast would help clarify the advantages of end-to-end co-optimization over staged approaches. Also a comparison with Self-Instruct (Wang et al., 2022) would strengthen the paper’s positioning. While both integrate prompting and fine-tuning.

Generating both prompts and LoRA parameters for each query introduces latency and computational cost, which may limit scalability and deployment in real-time settings.

The supervised regularization relies on GPT-4o-generated expert prompts, creating a dependency on an external model. The impact of weaker or smaller dataset on performance is not analyzed.

Despite strong empirical results, the paper provides little theoretical explanation of why shared representations improve generalization.

**Questions:**

What are the computational and latency trade-offs of generating both prompts and LoRA parameters at inference time, and how could these be mitigated for large-scale or real-time use?

How sensitive is MetaTuner’s performance to the quality and size of the GPT-4o-generated expert dataset, and how does it generalize when trained with weaker or smaller teachers?

Do the learned meta-prompts exhibit shared structures or transferable patterns across related tasks?

How model-specific are the learned meta-prompts, would fine-tuning different base models yield similar or divergent prompt behaviors?

---

> ### Author Response · Authors · 2025-11-22
> **Rebuttal by Authors [W1]**
>
> **[W1] The paper could more clearly differentiate MetaTuner from sequential prompt–fine-tuning pipelines such as Chain-of-Thought distillation (Finlayson et al., 2025; Wei et al., 2022). A stronger conceptual contrast would help clarify the advantages of end-to-end co-optimization over staged approaches. Also a comparison with Self-Instruct (Wang et al., 2022) would strengthen the paper’s positioning. While both integrate prompting and fine-tuning.**
>
> We sincerely thank the reviewer for the insightful comment. We believe our joint optimization framework has the following advantages over sequential methods when integrating prompt optimization and fine tuning:
>
> 1. **Task synergy by parameter sharing**.  Our joint optimization framework conducts prompt optimization and fine-tuning simultaneously, allowing gradients from different methods to interact within each optimization step. Since both prompt optimization and fine tuning aim to improve the answer quality of given questions, their gradients are often complementary, enabling the model to learn shared representations that benefit all tasks. In contrast, sequential optimization conducts prompt optimization and fine tuning in isolation; thus cannot reinforce each other.
> 2. **Avoids catastrophic forgetting.** In sequential optimization, prompt generation is trained first and then frozen, while the model is later fine-tuned on downstream tasks. This decoupling can cause the fine-tuning stage to overwrite or distort the knowledge in the prompts, leading to misalignment and catastrophic forgetting. MetaTuner avoids this by co-optimizing prompt and parameter generation end-to-end, keeping both components aligned and enabling them to adapt together throughout training—preserving task-relevant knowledge and improving robustness.
> 3. **Improved generalization.** Unlike sequential methods that optimize prompt generation and parameter adaptation separately, MetaTuner adopts a multi-task learning (MTL) perspective by jointly optimizing prompts and parameters through a shared meta encoder. The multi-task setup encourages the model to learn task-invariant features, constrains the hypothesis space, and acts as an implicit regularizer. As a result, MetaTuner generalizes better to unseen and out-of-distribution tasks than staged approaches.
>
> To further alliviate your concern, **we conduct experiments to compare our framework with Chain-of-Thought Distillation method and Self-Instruct.** For the CoT Distillation, we use GPT-4o to generate expert chain-of-thought data for supervised fine-tuning, while for the Self-Instruct method, we first expand the task instructions, then randomly concatenate it with each data instance to form the training input. From the results of Qwen2.5-3B across four datasets in the following table, **we can see our framework can indeed achieve better performance.**
>
> | Method           | MATH  | GSM8K | HotpotQA | CosmosQA |
> | ---------------- | ----- | ----- | -------- | -------- |
> | MetaTuner        | 41.33 | 73.08 | 58.85    | 87.15    |
> | CoT Distillation | 22.67 | 54.67 | 53.35    | 75.60    |
> | Self-Instruct    | 23.11 | 56.03 | 34.80    | 79.05    |

---

> ### Author Response · Authors · 2025-11-22
> **Rebuttal by Authors [W2 & Q1]**
>
> **[W2 & Q1] Generating both prompts and** **LoRA** **parameters for each** **query** **introduces latency and computational cost, which may limit scalability and deployment in** **real-time** **settings. What are the computational and latency trade-offs of generating both prompts and LoRA parameters at inference time, and how could these be mitigated for large-scale or real-time use?**
>
> Thanks for raising the important concern regarding inference latency and computational overhead.
>
> 1. **The computational cost of generating both prompts and LoRA parameters.**
>
> In the following, we report the cost of generating prompts and LoRA parameters, respectively. For reference, we also present the original inference cost without generating prompts and LoRA parameters, as well as the average best performance for each category of methods.
>
> - Average inference cost of Qwen2.5-3B base model per query (second).
>
> | Dataset          | MATH         | GSM8K        | HotpotQA     | CosmosQA     | Avg Cost     | Avg Best Performance |
> | -------------------- | ------------ | ------------ | ------------ | ------------ | ------------ | -------------------- |
> | Answer Generation    | 0.43         | 0.39         | 0.07         | 0.06         | 0.24         | 52.38               |
> | + Prompt             | 0.6 (+0.17)  | 0.55 (+0.16) | 0.23 (+0.16) | 0.18 (+0.12) | 0.39 (+0.15) | 57.57               |
> | + Prompt & Parameter | 0.73 (+0.30) | 0.66 (+0.27) | 0.25 (+0.18) | 0.48 (+0.42) | 0.53 (+0.29) | 65.10               |
>
> - Average inference cost of Qwen2.5-7B base model per query (second).
>
> | Dataset          | MATH         | GSM8K        | HotpotQA     | CosmosQA     | Avg Cost     | Avg Best Performance |
> | -------------------- | ------------ | ------------ | ------------ | ------------ | ------------ | -------------------- |
> | Answer Generation    | 0.71         | 0.68         | 0.14         | 0.11         | 0.41         | 60.30               |
> | + Prompt             | 0.99 (+0.28) | 0.95 (+0.27) | 0.44 (+0.3)  | 0.36 (+0.25) | 0.69 (+0.28) | 64.68               |
> | + Prompt & Parameter | 1.48 (+0.77) | 1.48 (+0.80) | 0.47 (+0.33) | 0.48 (+0.37) | 0.98 (+0.57) | 68.60               |
>
> From the above results, we can see,
>
> - Compared to methods with prompt generation, MetaTuner achieves an absolute performance gain of 5.73% at the cost of an average additional inference latency of 0.22 seconds.
> - Compared to Vanilla and Fine-tuning methods, MetaTuner achieves an absolute performance gain of 10.51% at the cost of an average additional inference latency of 0.43 seconds.
>
> 2. **The method to improve model inference efficiency.**
>
> To accelerate the inference process, **we further propose MetaTuner-C, which** **directly clusters the queries in the test set and then uses the prompt and parameters associated with the cluster centroid for all samples within that cluster during inference**. Assuming the test set contains $N$ samples and is partitioned into $K$ clusters, the additional time overhead of MetaTuner-C compared to vanilla methods can be reduced to $t=K(t_p+t_q)/N$, where $t_p$ and $t_q$ denote the time costs for generating the prompt and parameters for a single query, respectively. Notably, our main method, MetaTuner-J, can be viewed as a special case of MetaTuner-C when $K=N$. We evaluate MetaTuner-C with various $K$ using the Qwen2.5-3B base model on the MATH dataset. Our analysis shows that MetaTuner-C achieves strong performance while significantly reducing inference time. For example, under the Qwen2.5-3B setting, using the best fine-tuning baseline DPO as a reference, **MetaTuner-C with K=50 attains an absolute performance gain of 3.41% with only 0.03 seconds of additional latency per query**.
>
> - Performance of MetaTuner-C with Qwen2.5-3B model on MATH.
>
> | Method         | MetaTuner-J | MetaTuner-C (K=50) | MetaTuner-C (K=30) | MetaTuner-C (K=10) | DPO   |
> | -------------- | ----------- | ------------------ | ------------------ | ------------------ | ----- |
> | Performance    | 41.33       | 40.00              | 39.11              | 37.56              | 36.59 |
> | Inference Cost | 0.73(+0.3)  | 0.46(+0.03)        | 0.45(+0.02)        | 0.437(+0.007)      | 0.43  |
>
> - Performance of MetaTuner-C with Qwen2.5-7B model on MATH.
>
> | Method         | MetaTuner-J | MetaTuner-C (K=50) | MetaTuner-C (K=30) | MetaTuner-C (K=10) | DPO   |
> | -------------- | ----------- | ------------------ | ------------------ | ------------------ | ----- |
> | Performance    | 48.67       | 45.56              | 45.11              | 44.89              | 43.78 |
> | Inference Cost | 1.48(+0.77) | 0.80(+0.09)        | 0.76(+0.05)        | 0.73(+0.02)        | 0.71  |

---

> ### Author Response · Authors · 2025-11-22
> **Rebuttal by Authors [W3 & Q2, W4]**
>
> **[W3 & Q2] The supervised** **regularization** **relies on GPT-4o-generated expert prompts, creating a dependency on an external model. The impact of weaker or smaller dataset on performance is not analyzed. How sensitive is MetaTuner’s performance to the quality and size of the GPT-4o-generated expert dataset, and how does it generalize when trained with weaker or smaller teachers?**
>
> Thanks for the thoughtful question.
>
> **First,** we note that MetaTuner’s training consists of two phases: the warm-up phase and the joint training phase.
>
> - **During the warm-up phase**, we **train the prompt generator using expert data generated by GPT-4o**, enabling the model to develop strong instruction-following capabilities for prompt generation.
> - **During the joint training phase**, we do not use GPT-4o for data collection, instead we obtain the training data based on **the** **trained** **policy** **model**. A detailed analysis is provided in Section 4.4.
>
> **Second,** to further investigate the impact of GPT-4o–generated expert data during the warm-up stage, we additionally study how using a **weaker expert model** or a **smaller expert dataset** affects MetaTuner’s final performance. Below are the experimental results on the MATH dataset using the Qwen2.5-3B base model.
>
> - Performance of Qwen2.5-3B on MATH with smaller expert datasets.
>
> | Data Quatity                     | 100%  | 60%   | 20%   |
> | -------------------------------- | ----- | ----- | ----- |
> | Performance After Warmup         | 35.33 | 36.67 | 34.44 |
> | Performance After Joint Training | 41.33 | 41.33 | 40.89 |
>
> - Performance of Qwen2.5-3B on MATH with weaker expert datasets.
>
> | Expert Model                     | GPT-4o | Qwen2.5-32B | Qwen2.5-14B |
> | -------------------------------- | ------ | ----------- | ----------- |
> | Performance After Warmup         | 35.33  | 33.11       | 31.33       |
> | Performance After Joint Training | 41.33  | 40.22       | 40.00       |
>
> Our analysis reveals that MetaTuner exhibits robust performance under both weaker and smaller expert dataset settings:
>
> - As the size of the expert dataset decreases, performance remains stable both after warm-up and joint training.
> - When a weaker expert model is used for warm-up, the post-warm-up performance degrades moderately as the expert model’s capability diminishes. However, this performance gap is largely mitigated after joint training.
>
> This suggests that the warm-up phase only requires a basic level of prompt-generation competence. The proposed MetaTuner framework can subsequently discover superior prompt–parameter combinations during joint training phase, thereby enhancing overall performance.
>
>
>
> **[W4] Despite strong empirical results, the paper provides little theoretical explanation of why shared representations improve generalization.**
>
> We sincerely thank the reviewer for raising this insightful question regarding the theoretical motivation for generalization.
>
> Theoretically, the generalization capability of our framework stems from the representation sharing mechanism. In MetaTuner, we can regard Prompt Generation and Parameter Generation as two distinct tasks and enforce them to share a common meta encoder. **This shared-private multi-head architecture inherently induces a regularization effect [1,2]**: the shared encoder must simultaneously support the learning objectives of multiple heads, compelling it to capture stable, core structural features that are invariant across tasks and independent of the specific training distribution. **This aligns with classical theoretical results in multi-task learning (MTL) [3,4,5,6]:**
>
> > When multiple tasks share intermediate representations, the effective hypothesis space of the model is constrained, leading to tighter generalization error bounds.
>
> Intuitively, the model cannot learn highly task-specific, high-capacity representations independently for each task; instead, **all tasks must find compatible representations within the same shared feature space. This capacity constraint significantly reduces overfitting to the idiosyncrasies of the training data distribution, thereby enhancing generalization across tasks and datasets.** We believe that existing theoretical works on multi-task learning provide a solid foundation for understanding the generalization properties of MetaTuner.
>
> [1] Regularizing Neural Networks via Stochastic Branch Layers
>
> [2] Can Small Heads Help? Understanding and Improving Multi-Task Generalization
>
> [3] A Model of Inductive Bias Learning
>
> [4] The Benefit of Multitask Representation Learning
>
> [5] SHARING KNOWLEDGE IN MULTI-TASK DEEP REINFORCEMENT LEARNING
>
> [6] Multitask Learning via Shared Features: Algorithms and Hardness

---

> ### Author Response · Authors · 2025-11-22
> **Rebuttal by Authors [Q3, Q4]**
>
> **[Q3] Do the learned meta-prompts exhibit shared structures or transferable patterns across related tasks?**
>
> Thanks for the insightful comments. **Here, we provide several cases** (see the Table 1 in the link https://drive.google.com/file/d/1aLg0pnUQKvl9trNTfuOB9aMCQkKV5Bee/view?usp=sharing ）from the Qwen2.5-7B base model across four datasets to illustrate the phenomenon. From the examples, MetaTuner’s learned prompts exhibit clear cross-task commonalities and transferable patterns:
>
> - In MATH and GSM8K, both learned prompts refine the initial CoT instruction by explicitly guiding step decomposition, intermediate computation, and clear explanation—despite differing problem types.
> - In HotpotQA, the prompt preserves the core principle of concise, evidence-based answering, and sharpened to extract only task-relevant details while avoiding redundancy.
> - In CosmosQA, the prompt emphasizes grounding the answer in contextual key details.
>
> In summary, despite differences in task-specific requirements, **the prompts learned by MetaTuner consistently promote clear, logically structured responses and adapt their instructional details according to the nature of each task.** This indicates that MetaTuner is capable of identifying and leveraging shared structural patterns across tasks, while tailoring guidance to individual task needs—thereby enhancing the transferability and adaptability of its prompts across diverse settings.
>
>
>
> **[Q4] How model-specific are the learned meta-prompts, would fine-tuning different base models yield similar or divergent prompt behaviors?**
>
> Thanks for the insightful comments. Below, **we provide several examples** (see the Table 2 in the link https://drive.google.com/file/d/1aLg0pnUQKvl9trNTfuOB9aMCQkKV5Bee/view?usp=sharing ) from the MATH dataset to illustrate this phenomenon. As shown in the examples, the prompts learned by different base models (e.g., Qwen2.5-3B and Qwen2.5-7B) on the same tasks exhibit both commonalities and differences attributable to model capacity:
>
> - **Commonalities**: Both models **retain the core structure of the initial prompt** (e.g., “step-by-step reasoning,” “answer in \boxed{}”, and “word limit”), indicating that MetaTuner can stably optimize instruction formats that respect task constraints across different models.
> - **Differences**: **The 7B model tends to generate more task-specific and procedurally explicit instructions** (e.g., for geometry problems, explicitly instructing “first compute the new dimensions, then calculate the area change”), whereas **the 3B model favors more general guidance** (e.g., “explain the relevant mathematical concepts” or “include a table or diagram”).
>
> We hypothesize that **the difference is primarily due to the varying task-solving capabilities of the base models.** For the weaker 3B model, overly specific instructions risk embedding subtle errors (e.g., incorrect algebraic or geometric reasoning) that can mislead inference. MetaTuner thus learns to generate more robust, general prompts for smaller models. In contrast, the more capable 7B model can reliably follow and benefit from detailed, task-specific guidance, leading MetaTuner to produce finer-grained instructions for it.

---

### Author Response · Authors · 2025-12-01
**Summary of the Rebuttal**

We are thankful for the reviewers' thoughtful feedback and insightful comments. Below, we provide a concise summary of the rebuttal:

We propose MetaTuner, a framework featuring a shared meta-encoder and regularized supervision loss for joint prompt–parameter optimization in LLMs. Extensive experiments demonstrate its effectiveness and generalization, with all four reviewers recognizing the paper’s strengths, including:

- **Problem importance**: Reviewers `6Sni` and `LrBX` highlighted the value of unifying prompt learning and fine-tuning for principled task adaptation.
- **Elegant design**: Reviewer `6Sni` noted the the dual-decoder design with a shared meta-encoder; `ttpV` and `LrBX` praised the method’s clarity and intuitiveness.
- **Comprehensive experiments**: Reviewers `RbJU` and `ttpV` commended strong, consistent gains across datasets and models; `6Sni` appreciated thorough ablations and generalization analysis.
- **Writing quality**: All reviewers found the paper well-written and easy to follow.

While the reviewers recognized the above strengths, they also raised some concerns:

- **Paper positioning**: Reviewer `6Sni`urged clearer differentiation from staged methods like CoT distillation and Self-Instruct; `LrBX` called for sharper distinction from prior work such as BetterTogether.
- **Design choices and justification**: Reviewer `ttpV` questioned if gains come from joint training or increased parameters; `LrBX` challenged the use of discrete over soft prompts.
- **Generalization, robustness, and efficiency**:
  - **Generalization**: Reviewer `6Sni` called for theoretical justification and analysis of prompt transferability across tasks and models; `ttpV` raised concerns about catastrophic forgetting and requested more OOD evaluations.
  - **Robustness**: Both Reviewers `6Sni` and `LrBX` questioned sensitivity to the quality/size of GPT-4o–generated expert prompts, with `LrBX` also noting the lack of convergence analysis.
  - **Efficiency**: Reviewer `6Sni` highlighted inference latency and requested cost analysis and mitigation strategies; `RbJU` noted the loss of prompt optimization’s infrastructure simplicity and asked for training cost comparisons.
- **Experimental scope and presentation**: Reviewer `RbJU` suggested full fine-tuning and flagged minor writing issues; `LrBX` called for broader model and domain coverage.

During the discussion phase, we carefully addressed each reviewer's concerns. Below is a summary of the revised manuscript:

- **Theoretical** **grounding**: Provided a multi-task learning–based theoretical justification for MetaTuner’s improved generalization via shared representations (`6Sni`).
- **Experimental** **validation**:
  - Compared with CoT distillation, Self-Instruct (`6Sni`), and BetterTogether (`LrBX`), highlighting advantages over staged joint optimization approaches.
  - Confirmed gains arise from joint co-optimization architecture instead of parameter scaling via ablation and larger-model baselines (`ttpV`).
  - Showed discrete prompts outperform soft prompts in robustness for co-optimization (`LrBX`).
  - Extended evaluation to more tasks, models (`LrBX`), and full fine-tuning (`RbJU`); added OOD tests (`ttpV`) and case studies (`6Sni`) on learned prompt structures.
- **Robustness and stability**: Analyzed sensitivity to expert prompt quality/size (`6Sni`,`LrBX`) and provided convergence evidence via loss curves (`LrBX`).
- **Efficiency and deployment**: Reported training (`RbJU`)/inference (`6Sni`) costs; introduced MetaTuner-C with clustering-based fast inference (`6Sni`) and MetaTuner-P in prompt-only mode  (`RbJU`) to address latency and infrastructure concerns.
- **Clarity and presentation**: Added a comparative table vs. prior work (`LrBX`) and fixed minor writing issues (`RbJU`).

Eventually, before the software issue on Nov 27,

- **(Nov 23, 2025, 21:28 AoE) Reviewer `ttpV` raised the rating from 4 to 6**, acknowledging our theoretical and experimental validation of generalization and evidence that gains come from joint training instead of parameter scaling.
- **(Nov 25, 2025, 17:46** **AoE**) **Reviewer** **`LrBX`** **increased the rating from 2 to 4**, acknowledging broader evaluations, justification of discrete prompts, and robustness/convergence analyses. We also addressed further concerns on novelty with a comparative table and additional results, but score adjustment was no longer permitted due to the new rebuttal policy.
- **Reviewers `6Sni` and `RbJU` maintained their original positive ratings of 8**, and we thoroughly addressed their concerns in the rebuttal.

Below is the rating summary before and after rebuttal:

|Reviewer|6Sni|ttpV|RbJU|LrBX|Avg|
|--------------|----|----|----|----|----|
|Initial Rating|8|4|8|2|5.5|
|Updated Rating|8|6|8|4|6.5|

We sincerely thank the reviewers and ACs for their constructive feedback and guidance. Hope this summary will assist in making the final decision.

---

### Meta-Review · Area_Chair_5KK9 · 2026-01-09

**Summary:**

This submission proposes MetaTuner, an end-to-end framework that jointly optimizes prompts and lightweight parameter updates via a shared meta-encoder with dual decoders for post-training, aiming to unify prompt optimization and fine-tuning with co-optimization. Reviews were broadly positive on clarity and empirical strength, with key concerns centered on (i) generalization to unseen datasets and model families, (ii) whether gains are merely from additional parameters, and (iii) dependence on GPT-generated expert prompts for warm-up. The authors provided substantial additional evidence in the rebuttal and discussion that directly addresses these concerns. The method is clearly presented, and the empirical case is strengthened by broader evaluation and careful ablations. Therefore, the AC recommends Accept (Poster).

**Reviewer Concerns:**

Addressed concerns:
- Generalization to unseen datasets and model families: New generalization evaluations to unseen datasets and on additional model families were provided, consistently strengthening the paper’s empirical position
- Justification beyond increased parameters: Validated through comparisons with larger models, showing at least some gains are architectural rather than due to scaling
- Reliance on GPT-generated expert prompts for warm-up: New studies with smaller and weaker expert datasets demonstrate robustness and reduced dependence
- Conceptual contrast with prior literature: Clearer discussion and comparative analysis distinguishing MetaTuner from staged and related approaches
- Full fine-tuning: New experiments including full fine-tuning help clarify headroom and confirm that gains are not tied to LoRA alone.

Outstanding concerns:
- System complexity for training and serving: The method introduces additional complexity compared to pure prompt optimization. However, consistent with Reviewer RbJU, the AC agrees that the demonstrated gains and comprehensive evaluation compensate for this limitation, while noting that reducing complexity is an interesting and valuable direction for future work.

**Reviewer Scores:**

Given that the major concerns were addressed during rebuttal and discussion, the AC expects the positive reviews to maintain their scores, while the initial negative reviews would likely increase their ratings.

---

### Decision · Program_Chairs · 2026-01-26

Accept (Poster)